# Resveratrol Alleviated Oxidative Damage of Bovine Mammary Epithelial Cells via Activating SIRT5-IDH2 Axis

**DOI:** 10.3390/antiox14101171

**Published:** 2025-09-26

**Authors:** Hanlin Yang, Luya Liu, Xinyi Zhang, Shikai Gao, Anqi Li, Jinru Dong, Guangyang Lu, Qilong Yang, Xiaoxiao Liu, Shiang Sun, Heping Li, Yang Liu, Yueying Wang, Yingqian Han

**Affiliations:** 1Key Laboratory of Animal Biochemistry and Nutrition, Ministry of Agriculture and Rural Affairs, College of Veterinary Medicine, Henan Agricultural University, Zhengzhou 450046, China; lin13090428@163.com (H.Y.); 15038551056@163.com (L.L.); 13937691752@163.com (X.Z.); 18838038695@163.com (S.G.); 18237188802@163.com (A.L.); 17303725052@163.com (J.D.); 15890930310@163.com (G.L.); yqlhuiyi@163.com (Q.Y.); 15290686332@163.com (X.L.); 17855028248@163.com (S.S.); liheping@henau.edu.cn (H.L.); liuyang@henau.edu.cn (Y.L.); 2Key Laboratory of Veterinary Biotechnology of Henan Province, College of Veterinary Medicine, Henan Agricultural University, Zhengzhou 450046, China

**Keywords:** resveratrol, oxidative damage, bovine mammary epithelial cells, SIRT5, IDH2, succinylation

## Abstract

Effective intervention on oxidative damage of bovine mammary epithelial cells (bMECs) is particularly important for reducing the incidence rate of mastitis. As a natural antioxidant, resveratrol (RES) can scavenge ROS, protecting cells from oxidative damage. However, the role of RES in bMECs and its potential protective mechanism have not been fully elucidated. Our results indicated that RES alleviated oxidative damage and enhanced antioxidant capacity in bMECs. Furthermore, RES increased SIRT5 expression and interacted with SIRT5, which attenuated cellular oxidative stress, inflammatory response and autophagy activity. Interestingly, SIRT5 and RES further improved mitochondrial dysfunction by increasing intracellular NADPH and GSH levels. Meanwhile, RES activated SIRT5 to regulate enzymatic activity of SDH and IDH2, which were important enzymes for producing intracellular ATP and antioxidants. RES specifically activated SIRT5 to attenuate the succinylation levels of intracellular IDH2 associated with interacting with SIRT5. Collectively, these outcomes revealed that RES might function as an activator of SIRT5 to attenuate oxidative damage of bMECs via activating SIRT5-IDH2 axis, resulting in increased GSH and NADPH production. Therefore, RES may be useful to prevent and control bovine mastitis by relieving oxidative damage.

## 1. Introduction

Cows in early lactation or with high milk yield are usually subjected to metabolic disorders, with an increase in high energy and oxygen demand, causing reactive oxygen species (ROS) accumulation and inflammatory response, resulting in a decrease in milk production and economic harm to the dairy industry [1,2,3]. The imbalance between antioxidant function and ROS production caused by the hyper-metabolic mammary gland will increase the incidence of mastitis [4]. Mastitis is closely related to oxidative damage in the mammary gland, which is usually associated with a high ROS level and decreased antioxidant capacity [5]. Therefore, effective intervention regarding oxidative damage on bovine mammary epithelial cells (bMECs) is becoming increasingly important.

Mitochondria play an important role in endogenous ROS production and oxidative damage in cells, and mitochondrial dysfunction can cause cellular damage [6,7]. Sirtuins (SIRTs) participate in biological processes associated with mitochondria that result in changes in energy or ROS levels, which are responsible for various physiological processes [8,9]. SIRT5 belongs to the SIRT family, located in the mitochondrial matrix, cytoplasm, and nucleus, and has been described as playing crucial role in regulating oxidative stress, cellular metabolism, energy production, and ROS detoxification [10,11,12]. Interestingly, the data suggest that reducing oxidative damage is a possible mechanism by which SIRT5 protects various cells from apoptosis [13,14]. Under certain cellular stress conditions, SIRT5 exerts a pro-proliferative effect associated with pro-autophagy regulation [15]. In several cancers, such as colorectal cancer, gastric cancer, and osteosarcoma, SIRT5 acts as a proliferative factor to enhance autophagy action [15,16,17]. Overexpression of SIRT5 activates autophagy to delay the development of Alzheimer’s disease [18]. Reduced SIRT5 expression increases mitochondrial dysfunction [19]. SIRT5 indirectly regulates antioxidant glutathione (GSH) content and ROS generation induced by ammonia [20]. Our recent published article demonstrates that SIRT5 alleviates oxidative damage in bMECs exposed to ammonia [21]. Overall, SIRT5 plays a key role in resisting oxidative damage of the cell.

As a natural polyphenolic compound, resveratrol (RES) has many biological functions, such as antioxidant, anti-inflammatory, anti-aging, and anti-apoptotic effects, which have attracted much attention in the fields of medicine and biology [22,23,24,25]. RES can directly remove excessive ROS and elevate antioxidant capacity by upregulating antioxidant enzymatic activity, thereby reducing cell damage and maintaining intracellular redox balance [20,26,27,28]. Interestingly, RES is also the first discovered promising SIRT-activating compound [28,29]. Many studies have reported that RES alters SIRT activity [12], specifically activating SIRT1 to reduce ROS production and enhance antioxidant enzymatic activity in various types of cells [29,30,31]. In addition, studies have shown that RES has a strong ability to activate SIRT5 [12,32]. In the hippocampus of elderly rats, RES increases SOD1 and SIRT5 expression, and decreases NAD^+^ levels [28].

Currently, the use of bioactive substances to enhance breast defense mechanisms has become a research hotspot for preventing and curing mastitis. However, the role and potential protective mechanism of RES in bMECs have not been fully elucidated. In this paper, we hypothesized that RES reduced the oxidative damage of bMECs via activating SIRT5. Expectedly, RES alleviated oxidative stress and enhanced cellular antioxidant capacity. Furthermore, RES increased SIRT5 expression and interacted with SIRT5, which attenuated cellular oxidative stress, inflammatory response, and autophagy activity. Interestingly, SIRT5 and RES further improved mitochondrial dysfunction through increasing intracellular NADPH (nicotinamide adenine dinucleotide phosphate) and GSH levels. Mitochondrial dysfunction can impact ATP synthesis. Succinate dehydrogenase (SDH) is an indicator of mitochondrial function in ATP production, and mitochondrial destruction reduces SDH activity [33,34]. We found that SIRT5 and RES effectively enhanced the content of ATP and the enzymatic activity of SDH. Isocitrate dehydrogenase 2 (IDH2) plays a crucial role in the antioxidant pathway of mitochondria, causing NADPH and GSH production, which are the main antioxidants for improving ROS damage [35,36]. Our recent study demonstrates that SIRT5 enhances IDH2 activity through desuccinylation, increasing NADPH level to alleviate ammonia-induced ROS in bMECs [21]. Hence, we detected the level of IDH2 succinylation and found that RES enhanced the enzymatic activity of IDH2 via activating the SIRT5-IDH2 axis, resulting in increased GSH and NADPH production in bMECs exposed to H_2_O_2_. Collectively, RES improves mitochondrial dysfunction associated with activating SIRT5, and RES might act as an activator of SIRT5 to improve oxidative stress in bMECs via activating SIRT5-IDH2 axis, resulting in increased GSH and NADPH production. Therefore, RES may be useful to prevent and control bovine mastitis by relieving oxidative damage. These results not only offer a new perspective for the antioxidant defense role of RES in cells, but also provide theoretical evidence for antioxidant therapy strategies based on SIRT5.

## 2. Materials and Methods

### 2.1. Reagents and Antibodies

RES (Resveratrol, #501-36-0) was bought from TCI (Shanghai) Chemical Industry Development Co., Ltd. (Shanghai, China), NAM (nicotinamide, #HY-B0150), anti-flag magnetic beads (#HY-K0207), protein A/G magnetic beads (HY-K0202) and 3 × Flag peptide (#HY-P0319) were bought from MedChemExpress LLC (Monmouth Junction, NJ, USA). FBS (fetal bovine serum, #FSP500) was purchased from Shanghai ExCell biotechnology Co., Ltd. (Shanghai, China). Trypsin ethylenediamine tetraacetic acid digestion solution (GIBCO, New York, NY, USA, #25200072), high-sugar DMEM medium (HyClone, Logan, UT, USA, #SH30022.01), and PBS (phosphate-buffered saline, HyClone, #SH30256.01) were purchased from Cytiva, Inc. (Marlborough, MA, USA) RNAiso plus (#9109), real-time quantitative PCR kit (#RR820A) and reverse transcription kit (#RR047A) were from Takara Biotechnology Co., Ltd. (Dalian, China) BCA kit (#BCA01) and 5 × SDS loading buffer were from Beijing Dingguo Changsheng Biotechnology Co., Ltd. (Beijing, China). RIPA lysis buffer was from Thermo Fisher Scientific, Inc. (Waltham, MA, USA). Cell proliferation toxicity (CCK-8, #CA1210) was bought from Beijing Solarbio technology Co., Ltd. (Beijing, China). LDH (lactate dehydrogenase) cytotoxicity assay kit (#C0016), MDA (malondialdehyde, #S0131S), CAT (catalase, #S0051) activity, SOD (superoxide dismutase, #S0101S) activity, GSH-Px (glutathione peroxidase, #S0057S) activity, T-AOC (total antioxidant capacity, #S0121), ATP (#S0026), mitochondrial superoxide assay kit (#S0061S), mitochondrial membrane potential assay kit with TMRE (#C2001S), mitochondrial membrane potential assay kit with JC-1 (#C2006), autophagy staining assay kit with MDC (#S3018S), Mito-tracker red CMXRos (#C1035), NADP^+^/NADPH (#S0179), GSH/GSSG (#S0053) and cell mitochondria isolation kit (#C3601) were bought from Beyotime Biotechnology (Shanghai) Co., Ltd. (Shanghai, China). ROS (reactive oxygen species, #CA1420), bovine IL-6 (# SEKB-0365) and IL-8 kits (#SEKB-0366), SDH (succinate dehydrogenase, #BC0955) activity assay kit, and IDH2 (isocitrate dehydrogenase, #BC0405) activity assay kit were from Beijing Solarbio Technology Co., Ltd. (Beijing, China).

SIRT5 antibody (#15122-1-AP, 1:1000), IDH2 antibody (#15932-1-AP, 1:1000), β-actin antibody (#20536-1-AP, 1:5000), rabbit IgG control polyclonal antibody (#30000-0-AP, 4.0 μg for 3.0 mg of total protein lysate), HRP-conjugated IgG fraction monoclonal mouse anti-rabbit IgG, light chain specificity (#SA00001-7L, 1:5000), HRP-conjugated recombinant rabbit anti-mouse IgG kappa light chain (#SA00001-19, 1:5000), and HRP-conjugated AffiniPure goat anti-rabbit IgG (H+L) (#SA00001-2, 1:5000) were bought from Proteintech Group, Inc. (Rosemont, IL, USA). DYKDDDDK-Tag (#M20008, 1:5000) and HA-Tag (#M20003, 1:5000) were purchased from Abmart Medical Technology (Shanghai) Co., Ltd. (Shanghai, China). Succinylated lysine antibody (#3089, 1:1000) was bought from Wuhan DIA AN biotechnology Co., Ltd. (Wuhan, China).

### 2.2. Cells and Cell Grouping

Bovine mammary epithelial cells (bMECs), human embryonic kidney cells 293 (HEK293T cells), SIRT5-overexpressed cells (bMECs transfected with a pEGFP-SIRT5 vector) and pCDNA3.1 plasmid were provided by our laboratory, which were cultured and transfected according to our previous methods [21,37,38,39,40]. In this study, we used H_2_O_2_ as an oxidative stress agent to evaluate the antioxidant role of RES in bMECs. H_2_O_2_ (an oxidative stress agent) was dissolved in DMEM. RES (an antioxidant) was dissolved in DMSO. Nicotinamide (NAM, an inhibitor of SIRT5) was dissolved in PBS. When bMECs and SIRT5-overexpressed cells were cultured reaching the confluent extent of 80~90%, and they were randomly divided into seven groups. bMECs in the CT group were seeded in a basal medium. bMECs in the H_2_O_2_ group were treated with 500 µM H_2_O_2_ for 24 h. bMECs in the RESH group were handled with 40 μM RES and 500 μM H_2_O_2_ for 24 h. bMECs in the RESNH group were handled with 40 µM RES, 50 µM NAM, and 500 µM H_2_O_2_ for 24 h. bMECs in the NH group were treated with 50 µM NAM and 500 µM H_2_O_2_ for 24 h. SIRT5-overexpressed cells in the SOH group were treated with 500 µM H_2_O_2_ for 24 h. SIRT5-overexpressed cells in the SONH group were handled with 50 µM NAM and 500 µM H_2_O_2_ for 24 h.

HEK293T cells were planted in culture dishes. When the cell confluence was about 80%, they were randomly distributed into seven groups. Each group had three independent replicates. Cells were transfected with plasmids and treated according to our recent document [21]. Cells in the Flag-IDH2 group, the Flag-IDH2+H_2_O_2_ group, the Flag-IDH2+H_2_O_2_+RES group, the Flag-IDH2+H_2_O_2_+RES+NAM group, and the Flag-IDH2+H_2_O_2_+NAM group were transfected with the Flag-IDH2 and pcDNA3.1 plasmid and treated with or without 500 µM H_2_O_2_. In addition, 40 μM RES or 50 μM NAM were added or not accordingly at 24 h before sample collection. The Flag-IDH2+HA-SIRT5+H_2_O_2_ group and the Flag-IDH2+HA-SIRT5+H_2_O_2_+NAM group were transfected with the Flag-IDH2 and HA-SIRT5 plasmids, and both 500 µM H_2_O_2_ with or without 50 μM NAM were added at 24 h before sample collection.

### 2.3. Detection of Cell Viability

The cell counting kit (CCK-8) was used to detect the cell viability. Cells were cultured in 96-well plates (10^4^ cells per well) for 10 h and then treated with indicated agents for 24 h. 10 μL CCK-8 agent was added to each well, and after incubation at 37 °C for 2 h, the absorbance at 450 nm was measured. According to the instructions, cell viability was counted.

### 2.4. LDH Release Rate Analysis

LDH presents stability in cells under normal conditions. After cell death caused by stimulation, the plasma membrane ruptures and LDH is rapidly released into the extracellular space. As an important indicator of cell membrane integrity, LDH release is detected to evaluate the number of dead cells. Briefly, the medium was mixed with the reaction mixture in a 96-well plate for 30 min at room temperature, a stopping solution was added to stop the reaction, and the absorbance at 490 nm was measured. The LDH release was calculated based on the instructions.

### 2.5. Evaluation of ROS Level and MDA Content

The ROS level was measured with the Reactive Oxygen Species assay kit. Cells were lysed and centrifuged for 5 min (4 °C, 500× *g*). Collected supernatant was centrifuged again for 10 min (4 °C, 10,000× *g*). The fluorescent dye DHE was employed to assay ROS production. After cells were washed with PBS, the fluorescent dye (2 μM DHE) was added to incubate at 37 °C for 30 min for loading DHE. Then, PBS was used to wash cells, which were collected to observe the fluorescence intensity (Ex.535 nm, Em.610 nm) with a SpectraMax M5 molecular device (Microplate Reader, San Jose, CA, USA). MDA content in cells was assayed according to the instructions of the kit (Beyotime, #S0131S).

### 2.6. Detection of Inflammatory Markers and SIRTs

Enzyme-linked immunosorbent assay kits were employed to assay the content of inflammatory markers (IL-6 and IL-8) according to the instructions of manufacturer. The supernatant of cells after precipitation was collected for analysis. Real-time fluorescence quantitative PCR was employed to evaluate inflammatory markers (IL-6 and IL-8) and SIRT (SIRT3, SIRT4, and SIRT5) mRNA expression. Total RNA was obtained using the TRIzol regent. The PCR assay of QuantStudio 6-Flex (Applied Biosystems, Carlsbad, CA, USA) was optimized using SYBR green premix (TaKaRa, Tokyo, Japan). Briefly, the total volume was 10 μL, containing 1 μL template, 5 μL SYBR green premix, 1 μL primer mixture, and 3 μL ddH_2_O. PCR reaction was as follows: 95 °C for 2 min, then 40 cycles were performed at 95 °C for 15 s, 60 °C for 20 s, and 72 °C for 20 s. All experiments were repeated thrice. The mRNA abundance of target genes was evaluated with the 2^−ΔΔCt^ method [17,33,34,35,36]. β-actin was adopted as an internal gene. Primers for the real-time quantitative PCR assay were listed in Table 1.

### 2.7. Western Blotting

After incubation, cold PBS was used to wash cells two times, and the total proteins from cells were separated in each group using RIPA lysis buffer. The level of proteins was determined using a BCA kit. Then, adding 5 × SDS loading buffer, the separated proteins were denatured at 99 °C for 10 min. The proteins (30 µg/lane) were injected into 10% SDS-PAGE gels and finally transferred to polyvinylidene fluoride (PVDF) membranes (EMD Millipore, Darmstadt, Germany). The membranes were then blocked for 1.5 h with 5% skim milk diluted in Tris-buffer saline containing Tween-20 (TBST) and then washed with TBST for 30 min. Next, the membranes were inoculated with the corresponding antibodies overnight at 4 °C, and TBST was employed to clean the membranes for 30 min. Finally, the membranes were inoculated with horseradish peroxidase (HRP)-conjugated anti-IgG antibody for 1.5 h at 20~25 °C. The protein bands were visualized under a developer using an enhanced chemiluminescence reagent (EMD Millipore), and the grayscale values of the protein bands were analyzed using ImageJ software 1.4.3.67 (National Institutes of Health, Bethesda, MD, USA).

### 2.8. Analysis of Antioxidant Markers

The treated cells were washed and collected using a cell scraper. Next, the cells were centrifuged and incubated with 150 μL of Western and IP lysis buffers along with protease inhibitors. The supernatant was then removed by centrifugation following extraction using a 1 mL sterile syringe 40 times. Afterward, the T-AOC content, the activity of CAT, SOD, and GSH-Px, as well as NADP^+^/NADPH and GSH/GSSG were assayed based on the kit manual.

### 2.9. Observation of Mitochondrial Morphology

Mito-Tracker Red CMXRos can specifically mark live mitochondria in cells. According to the instructions, cells were loaded with Mito-Tracker Red CMXRos at 37 °C for 15 min. Cells were then cleaned with PBS three times and observed under a confocal scanning microscope (LSM 800, Zeiss, Jena, Germany).

### 2.10. Measurement of Mitochondrial Membrane Potential

The mitochondrial membrane potentials (MMPs) were evaluated using mitochondrial membrane potential assay kits with TMRE or JC-1. JC-1 (an ideal fluorescent probe) is extensively employed for assaying MMP (∆Ψm). When MMP is high, JC-1 forms aggregates in the mitochondrial matrix, which emit red fluorescence. At low MMP, JC-1 is a monomer that emits green fluorescence. The ratio of green and red fluorescence intensity is usually adopted to evaluate the proportion of mitochondrial depolarization. Briefly, cells were treated with dye JC-1 for 20 min at 37 °C and the fluorescence was detected under a confocal scanning microscope (LSM 800, Zeiss, Germany) according to the manual. Due to the green fluorescence displayed in SIRT5-overexpressed cells (bMECs transfected with pEGFP-SIRT5 vector) under fluorescence microscopy, the JC-1 kit is not suitable for detecting the mitochondrial membrane potential of SIRT5-overexpressed cells. TMRE (an orange-red cationic fluorescent probe) penetrates the cell membrane to aggregate in intact mitochondria, which emit bright orange-red fluorescence. Under depolarization or inactive status, MMP declines and TMRE exists in the cytoplasm, and the fluorescence intensity of orange-red within the mitochondria is obviously weakened. Indicated cells were handled with the TMRE dye at 37 °C for 15 min, and the fluorescence was evaluated under a confocal scanning microscope (LSM 800, Zeiss, Germany) according to the manual.

### 2.11. Cellular Autophagy Vesicle Detection

MDC (Monodansylcadaverine, a fluorescent probe) is commonly used to detect the autophagy vesicle, which can mark autophagosomes (autophagosome) through specific binding to membrane lipids. According to the instructions, 1 mL MDC of staining solution was added into the cells, and the cells were kept in a dark place using a cell incubator at 37 °C for 30 min. A fluorescence microscope (IX73, Olympus Corporation, Tokyo, Japan) was used to observe the cells, which calculated the average fluorescence intensity using Image J software.

### 2.12. Transmission Electron Microscopy (TEM) Observation

Transmission electron microscopy observation is a common method for evaluating the formation of the autophagy vesicle. After cell treatment, the fixative solution was used to fix cells for 5 min at 20 °C. A cell scraper was used to gently collect the cells in one direction, and then the cells were kept from light at 20 °C for 30 min. At the end, cells were observed and photographed with TEM (HT7700, Hitachi, Tokyo, Japan).

### 2.13. Mitochondrial Superoxide Production

MitoSOX Red can quickly and specifically target mitochondria as a fluorescent probe for detecting selectively superoxide in mitochondria. The mitochondrial superoxide production was evaluated with MitoSO^TM^ Red fluorescence intensity according to the instructions (Beyotime, #S0061S). Indicated cells were mixed with MitoSO^TM^ Red and kept from light at 37 °C for 20 min. Next, PBS was used to clean cells three times, and the fluorescence intensity (Ex.535 nm, Em.610 nm) of cells was measured with the fluorescent enzyme-linked immunosorbent assay reader SpectraMax M5 (Molecular Device, San Jose, CA, USA).

### 2.14. Determination of ATP Content, IDH2 and SDH Enzymatic Activity

The ATP content of cells was detected using an ATP assay kit (Beyotime, #S0026) according to the manual. Mitochondrial fractions of cells were prepared using the cell mitochondria isolation kit (#C3601) by the manufacturer’s protocol. IDH2 and SDH activity was evaluated using the corresponding reagent kits (Solarbio, #BC0405; #BC0955) according to the manufacturer’s instructions. Detailed operational steps were conducted according to our recent literature [21].

### 2.15. Virtual Screening Based on Molecular Docking

The SDF format of small molecules (RES) was obtained from the PubChem database (https://pubchem.ncbi.nlm.nih.gov/, accessed on 15 January 2025) and transferred to mol2 files using the Open Babel software 3.1.1. Molecular structures of several proteins (three SIRTs, IDH2, and SDHA) were obtained from the Uniprot database (https://www.uniprot.org/, accessed on 15 January 2025). The molecular docking binding energy is used as an indicator to judge the interaction effect. Generally, the larger the absolute value of the molecular docking binding energy, the stronger the interaction effect between the ligand and the receptor. Binding energy values less than 0 kcal/mol indicate that there is an interaction between ligands and receptors. Moreover, binding energy values less than −5 kcal/mol indicate that there is a strong interaction between ligands and receptors. AutoDock Vina software 1.5.6 is used for the molecular docking of RES and several proteins to gain the binding energy values. The optimal binding conformation was visualized using PyMOL software 3.1.3 to show a 3D conformation map.

### 2.16. Interaction Networks Between IDH2 and SDHA with SIRT5

IDH2 had been validated in the interaction network of bovine SIRT5 [21]. We further imported SIRT5, IDH2, and SDHA, respectively, into the STRING database (https://cn.string-db.org/, accessed on 18 January 2025) and named the species Bos taurus. We then clicked it to search, downloaded, and saved the result as a bitmap image.

### 2.17. The Model of Interaction Between SIRT5 and SDHA

The docking data displayed that SIRT5 specifically formed hydrogen bonds with the K413 site of IDH2 and took shape the protein-protein interfaces with IDH2 [21]. SDH as a mitochondrial marker enzyme binds to the mitochondrial inner membrane of the eukaryote, which is a highly conserved heterotetrameric protein containing the four subunits SDHA/B/C/D, catalyzing the oxidation of succinic acid in the tricarboxylic acid (TCA) cycle [41]. SDHA catalyzes the conversion of succinic acid to fumaric acid while transferring electrons to produce FADH2, which is one of the reactions in the tricarboxylic acid cycle. SIRT5 was verified to interact with SDHA, and SIRT5 silencing caused hypersuccinylation and reactivation of SDHA [42]. Therefore, the interaction between SIRT5 and SDHA was modeled based on molecular docking methods. Molecular structures of SIRT5 and SDHA were obtained from the Uniprot database (https://www.uniprot.org/, accessed on 15 January 2025). The GRAMM docking web server (https://gramm.compbio.ku.edu/, accessed on 19 January 2025) was used to analog their interactions. The optimal binding conformation was visualized using PyMOL software.

### 2.18. Determination of IDH2 Succinylation Level

To measure the succinylation level of IDH2, HEK293T cells were transfected subsequently with HA-SIRT5 and Flag-IDH2-recombinant plasmids. The detailed operation was described in the literature [21].

### 2.19. Data Statistical Processing

All of the data were listed as mean ± SEM and processed using GraphPad Prism 6 software. Firstly, homogeneity of variance analysis was performed using F-test, then two-tailed unpaired *t*-test was executed between two groups, and finally, one-way ANOVA was performed among multiple groups. After adjusting post hoc Bonferroni to determine the difference, *p* < 0.05 was considered significant.

## 3. Results

### 3.1. Screening the Concentration of H_2_O_2_ and RES for Treating Cells

To confirm the appropriate concentration of H_2_O_2_ and RES for treating cells, we used the CCK-8 method to detect cell viability and combined with the result of LDH release. As shown in Appendix A, H_2_O_2_ reduced the viability of bMECs in a concentration-dependent manner. At a concentration of 500 μM, H_2_O_2_ significantly reduced cell viability to 50%. In Appendix A, low-concentration RES did not affect the viability of bMECs, while high-dosage RES obviously reduced the viability of bMECs in a concentration-dependent manner. Expectedly, RES (<50 µM) enhanced the viability of bMECs treated with 500 µM H_2_O_2_ in a dosage-dependent pattern (Appendix A). In addition, 500 μM H_2_O_2_ enhanced LDH release in the supernatant of cell culture (Appendix A). After RES treatment, LDH release in the RESH group was reduced markedly (Appendix A). Hence, we chose 500 μM H_2_O_2_ and 40 µM RES to challenge cells for the subsequent experiments.

### 3.2. SIRT5 Attenuated Cellular Oxidative Stress and Inflammatory Response

SIRT5 can reduce ROS levels and protect cells from oxidative damage, making it a potential target for managing oxidative stress-related conditions [12,21]. In this study, we used H_2_O_2_ to treat SIRT5-overexpressed cells with or without NAM (an inhibitor for SIRT5) treatment. Compared with the H_2_O_2_ group, as listed in Appendix A, the LDH release in SIRT5-overexpressed cells decreased markedly in the SOH group. In Figure 1A,B, the ROS level and MDA content in the SOH group were obviously lower than those in the H_2_O_2_ group. Moreover, the mRNA abundance and content of IL-6 and IL-8 in the SOH group were also reduced evidently compared with the H_2_O_2_ group (Figure 1C,F). Furthermore, the content of T-AOC and the enzymatic activity of SOD, GSH-Px, and CAT in the H_2_O_2_ group were obviously lower than those in the CT group (Figure 2). Conversely, the content of T-AOC and the enzymatic activity of SOD, GSH-Px, and CAT in the SOH group were higher than those in the H_2_O_2_ group (Figure 2). Inhibition of SIRT5 with NAM also confirmed that the LDH release, ROS level, and MDA content were increased significantly in the SONH group compared with the SOH group (Appendix A and Figure 1A,B). Similarly, the mRNA abundance and content of IL-6 and IL-8 in the SONH group were also significantly increased compared with the SOH group (Figure 1C,F). Consistently, compared with the SOH group, the content of T-AOC and the enzymatic activity of SOD, GSH-Px, and CAT in the SONH group were significantly increased (Figure 2). The above results indicated that SIRT5 attenuated cellular oxidative stress and inflammatory response and further enhanced the antioxidant capacity of cells.

### 3.3. RES Ameliorated Oxidative Stress and Enhanced Cellular Antioxidant Capacity Associated with SIRT5

ROS and MDA are commonly used as biomarkers of oxidative stress [25], and exposure to H_2_O_2_ induced an obvious elevation in the intracellular ROS level (Figure 3A) and MDA content (Figure 3B) of bMECs. Excess ROS production can cause inflammation [43]. Consistently, the mRNA abundance and content of IL-6 and IL-8 in the H_2_O_2_ group were elevated evidently compared with the CT group (Figure 3C,F). As expected, RES can alleviate the oxidative stress induced by H_2_O_2_. Compared with the H_2_O_2_ group, the ROS level and MDA content were significantly decreased in the RESH group (Figure 3A,B). In addition, the mRNA abundance and content of IL-6 and IL-8 in the RESH group were significantly decreased (Figure 3C,F). Collectively, 500 μM H_2_O_2_ induced the oxidative stress in bMECs, accompanied by the occurrence of cellular inflammatory response, and RES ameliorated the oxidative stress and inflammatory response induced by H_2_O_2_. The major antioxidant enzymes, containing SOD, CAT, and GSH-Px, prevent the production of the oxidation chain by scavenging molecules responsible for free radical production [44,45]. Interestingly, RES can directly remove excess ROS, enhance antioxidant capacity, reduce cell damage, and maintain intracellular redox balance by regulating antioxidant enzymatic activity (SOD, GSH-Px) [26,27,28]. Moreover, we found that 500 μM H_2_O_2_ markedly reduced the content of T-AOC and the enzymatic activity of SOD, GSH-Px, and CAT (Figure 4). In addition, in contrast to the H_2_O_2_ group, the content of T-AOC and the enzymatic activity of SOD, GSH-Px, and CAT were significantly increased in the RESH group (Figure 4). The changes in enzymatic activity indicated that H_2_O_2_ attenuated the antioxidant capacity of bMECs while RES elevated the antioxidant capacity of bMECs to further resist H_2_O_2_-induced oxidative stress.

To further demonstrate the correlation between RES reducing oxidative stress and SIRT5, we used H_2_O_2_ and RES to treat bMECs with or without NAM treatment. Compared with the H_2_O_2_ group, the ROS level and MDA content of bMECs in the NH group was significantly increased (Figure 3A,B). Interestingly, this was also the case in the RESNH group compared with the RESH group (Figure 3A,B). Similarly, compared with the H_2_O_2_ group, the mRNA abundance and content of IL-6 and IL-8 in the NH group were significantly increased (Figure 3C,F). Consistently, the same was true for the RESH group compared with the RESNH group (Figure 3C,F). Compared with the NH group, these indicators were obviously decreased in the RESNH group (Figure 3). Moreover, inhibiting endogenous SIRT5 can weaken the antioxidant role of RES. Similarly, the content of T-AOC and the enzymatic activity of SOD, GSH-Px, and CAT were significantly reduced in NH cells compared with the H_2_O_2_ group (Figure 4). Additionally, in contrast with the RESH group, the content of T-AOC and the enzymatic activity of SOD, GSH-Px, and CAT were significantly decreased in the RESNH group (Figure 4). Compared with the NH group, these indicators were obviously increased in the RESNH group (Figure 4). The above results further suggested that both RES and endogenous SIRT5 can reduce these indexes of the oxidative stress, implying that RES resisted oxidative stress associated with SIRT5.

### 3.4. RES Increased SIRT5 Expression and Interacted with SIRT5

There is a contradiction in activating SIRT5 with RES. Some studies indicate that RES has a potent ability to activate SIRT5 [32,46]. However, other reports suggest that RES elevates the desuccinylating activity of SIRT5 and has no effect on the expression of the SIRT5 protein [28] since three SIRTs (SIRT3, SIRT4, and SIRT5) distributed in the mitochondria participate in regulating the basic biology of mitochondria and ROS detoxification [6,10,11,12]. Firstly, we detected the mRNA abundance of SIRT3, SIRT4, and SIRT5. As listed in Figure 5A–C, the mRNA abundance of SIRT3 and SIRT5 in bMECs were increased significantly by 40 µM RES, while when the mRNA abundance of SIRT4 decreased, there was no significant difference (*p* > 0.05). Surprisingly, the level of the SIRT5 protein in bMECs was increased significantly by 40 µM RES (Figure 5D). Next, the molecular docking method was adopted to predict the interactions between RES and three SIRTs. The binding energy value was −7.7 kcal/mol (Figure 6A), −9.0 kcal/mol (Figure 6B), and −9.0 kcal/mol (Figure 6C) between RES and SIRT3 or SIRT4 or SIRT5, respectively. The highest binding energy for molecular docking was visualized using the PyMOL software, indicating that RES bound to hydrogen bonds in the active pocket of SIRT3 through PHE-116 and THR-191 (Figure 6A), in the active pocket of SIRT4 through TYR-75, ARG-76, GLY-261, and SER-263 (Figure 6B), as well as in the active pocket of SIRT5 through ASN-141 (Figure 6C). These data indicated that there was a strong interaction between RES and SIRT3 or SIRT4 or SIRT5. Specially, RES activated SIRT5 expression, and the binding energy value was −9.0 kcal/mol with SIRT5, suggesting that RES might alleviate oxidative stress via activating SIRT5.

### 3.5. RES Elevated NADPH and GSH Contents via Activating SIRT5

Mitochondrial dysfunction caused oxidative stress to increase the ratio of NADP^+^/NADPH [47,48], which led to high oxidized states of the glutathione redox couple (GSH/GSSG). As listed in Figure 7, cells exposed to H_2_O_2_ demonstrated a high NADP^+^/NADPH and low GSH/GSSG ratios contrast with cells in the CT group. The elevated NADP^+^/NADPH and declined GSH/GSSG ratios in cells of the H_2_O_2_ group were recovered by an overexpression of SIRT5. Compared with cells in the H_2_O_2_ group, there was an evident decline in the ratio of NADP^+^/NADPH and a huge enhancement in the ratio of GSH/GSSG in SIRT5-overexpressed cells treated with H_2_O_2_ (Figure 7A,B). In addition, the decreased NADP^+^/NADPH and increased GSH/GSSG ratios in cells of the SOH group was recovered by inhibiting SIRT5. Compared with cells in the SOH group, there was an obvious elevation in the ratio of NADP^+^/NADPH and an evident decline in the ratio of GSH/GSSG when SIRT5-overexpressed cells were treated with H_2_O_2_ and NAM co-treatment (Figure 7A,B). These suggested that SIRT5 elevated NADPH and GSH contents to alleviate cellular oxidative stress.

Next, the elevated NADP^+^/NADPH and decreased GSH/GSSG ratios in the H_2_O_2_ group were changed by RES. Compared with cells in the H_2_O_2_ group, there was an evident decline in the ratio of NADP^+^/NADPH and a huge enhancement in the ratio of GSH/GSSG when bMECs were treated with H_2_O_2_ and RES (Figure 7C,D). In addition, the decreased NADP^+^/NADPH and increased GSH/GSSG ratios in the RESH group was recovered by inhibiting SIRT5. Compared with cells in the RESH group, there was a substantial elevation in the NADP^+^/NADPH ratio and an obvious decline in the ratio of GSH/GSSG when cells in H_2_O_2_ and RES co-treatment were exposed to NAM (Figure 7C,D). Compared with the NH group, the ratios of NADP^+^/NADPH and GSH/GSSG were obviously reversed in the RESNH group (Figure 7). These results further suggested that RES enhanced NADPH and GSH contents to alleviate oxidative stress via activating SIRT5.

### 3.6. RES Improved Mitochondrial Dysfunction via Activating SIRT5

A large number of ROS will exhaust GSH to damage mitochondria and produce higher cellular ROS levels [49]. Hence, improving mitochondrial functional disorder could ameliorate ROS release and oxidative damage [50]. SIRT5 plays a crucial role in regulating mitochondrial function and metabolic homeostasis [51]. We detected mitochondrial morphology, MMP, autophagy vesicle, and superoxide production to explore the effect of SIRT5 or RES on mitochondrial dysfunction. As displayed in Figure 8, the mitochondrial morphology and structure in the CT group were normal and linear, and mitochondria swelling, breakage, and fragmentation were observed in the H_2_O_2_ group. Moreover, the mitochondrial morphology and structure in the SOH and RESH groups were similar to those in the CT group. As expected, inhibition of SIRT5 with NAM reduced the mitochondrial number and caused mitochondrial swelling and fragmentation in the NH group contrast with the H_2_O_2_ group (Figure 8). In the RESNH or SONH group, the mitochondrial morphology was obviously fragmented, compared with the RESH or SOH group, respectively (Figure 8). Expectedly, compared with the NH group, the morphology of mitochondria in the RESNH group was obviously linear (Figure 8).

In Figure 9A,B, compared to the CT group, the ratio of JC-1 monomer/aggregate in the H_2_O_2_ group was elevated. On the contrary, after co-treatment with RES and H_2_O_2_, there was an obvious decline in the RESH group. Inhibition of SIRT5 increased the ratio, which was higher in the NH and RESNH groups than in the H_2_O_2_ and RESH groups, respectively. Contrasting with the NH group, the ratio of JC-1 monomer/aggregate was obviously decreased in the RESNH group (Figure 9A,B). In Figure 9C,D, in the CT group, accumulated TMRE emitted bright orange-red fluorescence in the mitochondria. In the H_2_O_2_ group, the orange-red fluorescence intensity within the mitochondria was evidently weakened. In contrast, when SIRT5-overexpressed cells were exposed to H_2_O_2_, there was a stronger fluorescence in the mitochondria (SOH group). In the SONH group, there was a weaker fluorescence in the mitochondria than that in the SOH group.

As listed in Figure 10, massive autophagosomes and autolysosomes were present in the H_2_O_2_ group. Conversely, the number of autophagosomes and autolysosomes in the SOH group and RESH group were evidently reduced compared with the H_2_O_2_ group (Figure 10). Furthermore, MDC staining was used to mark and observe the autophagosome (Figure 11A). Expectedly, compared with the CT group, H_2_O_2_ induced an increased level of autophagy (Figure 11B,C). In contrast, the level of autophagy was obviously decreased in the SOH and RESH groups compared with the H_2_O_2_ group. Inhibition of SIRT5 with NAM resulted in a higher level of autophagy in the NH group, which was evidently more enhanced than that in the H_2_O_2_ group (Figure 11B). In the RESNH or SONH group, the level of autophagy was significantly higher than that in the RESH or SOH group (Figure 11B,C). The above results indicated that RES and SIRT5 inhibited the autophagy in bMECs induced by H_2_O_2_.

In Figure 12A,B, compared with the CT group, the level of mitochondrial superoxide in the H_2_O_2_ group was evidently enhanced. Conversely, the level of mitochondrial superoxide in the SOH and RESH groups was significantly decreased compared with the H_2_O_2_ group. Expectedly, there was an enhanced level of mitochondrial superoxide in the NH group through NAM inhibiting SIRT5, significantly higher than that in the H_2_O_2_ group (Figure 12B). In the SONH group, the mitochondrial superoxide production was significantly increased compared with the SOH group (Figure 12A). Similarly, the mitochondrial superoxide production of the RESNH group were significantly elevated in contrast with the RESH group. The mitochondrial superoxide production of the RESNH group was obviously decreased compared with the NH group (Figure 12B). In short, RES improved mitochondrial morphology, enhanced MMP, and reduced the level of autophagy and mitochondrial superoxide production associated with SIRT5.

Mitochondrial damage can mitigate the synthesis of ATP. As shown in Figure 12C,D, H_2_O_2_ markedly reduced ATP content, yet SIRT5 and RES evidently recovered this reduction, enhancing ATP content in the SOH and RESH groups. Furthermore, inhibiting endogenous SIRT5 using NAM, ATP content was again attenuated in the NH group and lower than that in the H_2_O_2_ group (Figure 12D). In the SONH and RESNH groups, ATP levels were lower than that in the SOH and RESH groups, respectively. Contrasting with the NH group, ATP content was undoubtedly elevated in the RESNH group (Figure 12D). SDH is a functional indicator of mitochondrial ATP production, and a decrease in SDH activity denotes mitochondrial damage [33,34]. As indicated in Figure 13A,B, H_2_O_2_ markedly inhibited SDH enzymatic activity, and overexpression of SIRT5 and RES noticeably obstructed this inhibition induced by H_2_O_2_. Additionally, inhibiting endogenous SIRT5 with NAM further decreased the SDH enzymatic activity (Figure 13A,B). Compared with the NH group, the SDH enzymatic activity was clearly elevated in the RESNH group (Figure 13B). IDH2 was required as a substrate of SIRT5 to ameliorate the oxidative stress induced by NH4Cl [21]. We speculated that RES attenuated the oxidative damage in bMECs by regulating IDH2 activity. As expected, in contrast with the CT group, H_2_O_2_ reduced IDH2 enzymatic activity (Figure 13C,D). Conversely, IDH2 enzymatic activity in the SOH and RESH groups was evidently enhanced in contrast with the H_2_O_2_ group. Expectedly, when inhibiting SIRT5 using NAM, there was lower IDH2 enzymatic activity in the NH group, and notably lower than that in the H_2_O_2_ group (Figure 13D). In the SONH group, the enzymatic activity of IDH2 evidently declined compared with the SOH group (Figure 13C). Similarly, the IDH2 enzymatic activity of the RESNH group was significantly declined in contrast with the RESH group, and the enzymatic activity of IDH2 of the RESNH group were evidently higher than that in the NH group (Figure 13D).

SDH and IDH2 were identified within the SIRT5 interaction network of cattle from the STRING database (https://cn.string-db.org/, accessed on 18 January 2025) (Figure 14A). To further explore the correlation between SIRT5 and SDH, we obtained the predicted Bos taurus SIRT5 and SDHA protein structures from the Uniprot database (https://www.uniprot.org/, accessed on 15 January 2025) and the GRAMM Docking Web Server (https://gramm.compbio.ku.edu/, accessed on 19 January 2025), which were utilized to simulate their possible interactions. The data verified that SIRT5 could form hydrogen bonds with SDHA at the K362 site (Figure 14B). Therefore, we speculate that K362 may be a crucial site for SIRT5-regulated post-translational modification, but this needs to be further demonstrated by mass spectrometry and other experimental methods. The binding energy value was −7.3 kcal/mol and −6.93 kcal/mol between RES and SDHA or IDH2, respectively (Figure 14C,D). The highest binding energy for molecular docking is visualized using PyMOL software, indicating that RES bound to hydrogen bonds in the active pocket of SDHA through LYS-129, ARG-594, and TYR-665 (Figure 14C), as well as in the active pocket of IDH2 through PRO-73 and ARG-393 (Figure 14D). These data suggested that there was a strong interaction between RES and SDHA or IDH2. Collectively, RES improved mitochondrial dysfunction via activating SIRT5.

### 3.7. RES Activated SIRT5-IDH2 Axis to Enhance IDH2 Activity

SIRT5 reduced intracellular IDH2 succinylation levels by interacting with endogenous IDH2, and further enhanced IDH2 enzymatic activity via desuccinylation modification, resulting in increased NADPH production [21]. In this study, RES enhanced IDH2 activity via activating SIRT5 (Figure 13C,D). Moreover, inhibiting SIRT5 with NAM also confirmed this point. Next, we detected the succinylation level of IDH2. As illustrated in Figure 15A,B, HEK293T cells co-transfected with Flag-IDH2 and HA-SIRT5 plasmid exposure to H_2_O_2_ led to an evident decline in the succinylation level of IDH2, which was restored using NAM treatment, compared with HEK293T cells transfected with Flag-IDH2 plasmid exposure to H_2_O_2_. Similarly, in Figure 15C,D, HEK293T cells transfected with Flag-IDH2 plasmid exposure to H_2_O_2_ and RES led to a notable reduction in the succinylation level of IDH2, which was restored upon NAM treatment, compared with HEK293T cells transfected with Flag-IDH2 vector exposure to H_2_O_2_. HEK293T cells transfected with Flag-IDH2 plasmid exposure to H_2_O_2_ and NAM led to an increasing trend in the IDH2 succinylation level compared with HEK293T cells transfected with Flag-IDH2 plasmid exposure to H_2_O_2_, further illustrating that inhibiting endogenous SIRT5 reduced the desuccinylation level of IDH2. Overall, these results verified that RES reduced intracellular succinylation levels of IDH2 by interacting with endogenous IDH2 and catalyzing its desuccinylation, further elevating IDH2 enzymatic activity via activating SIRT5. In this way, RES might act as an activator of SIRT5 and enhance IDH2 enzymatic activity via activating the SIRT5-IDH2 axis, resulting in increased GSH and NADPH production, improving oxidative damage in bMECs.

## 4. Discussion

At present, mastitis remains the most important disease that poses a threat to the development of the global dairy industry. It is widely recognized that oxidative stress plays an important role in bovine mastitis [52,53]. Recent evidence suggests the interaction between inflammation and oxidative stress [54]. Under a physiology environment, mitochondria elevate an antioxidant defense system to remove ROS and guarantee balanced redox homeostasis. Nevertheless, excess ROS production leads to mitochondrial dysfunction, triggering a variety of pathologies [55,56]. Hence, the maintenance or improvement of mitochondrial function may be a crucial way to cure diseases. The SIRT family participates in the processes of antioxidant and oxidative damage, such as mitochondrial metabolism and function and DNA damage repair [57]. In addition, NAD^+^ is necessary for the SIRT family to exert deacetylase activity. NAD^+^ is also a crucial molecule of redox signaling, which provides evidence that SIRTs, as an indispensable participant, regulate cellular antioxidant and redox signaling pathways [58]. SIRT5 exerts a key role in guaranteeing metabolic homeostasis, which regulates many molecules to elevate antioxidant defense in cells and maintains cellular redox balance by activating some enzymes, including superoxide dismutase 1 (SOD1), which exerts a key role in neutralizing ROS [14,51]. In addition, deletion of SIRT5 ameliorates SOD2 expression in mice [59]. In addition, SIRT5 can control the networks of mitochondrial metabolism to remove ROS and decrease mitochondrial oxidative damage [11,60,61], as verified with a decline in mitochondrial H_2_O_2_ and superoxide levels, as well as the restoration of mitochondrial morphology and MMP [62]. Therefore, SIRT5 has been considered an interesting therapeutic target for oxidative stress-regulated disease [58]. In this study, 500 µM H_2_O_2_ caused oxidative stress in bMECs, accompanying increased LDH release, ROS level, and MDA content, as well as further causing cellular inflammation with a high level of inflammatory markers. The indexes of oxidative damage and antioxidant capacity further demonstrated that SIRT5 reduced the oxidative damage caused by H_2_O_2_ and elevated the antioxidant capacity of cells, accompanied by low LDH release, ROS level, and MDA content, as well as a low level of inflammatory markers, while T-AOC, CAT, SOD, and GSH-Px enzyme activities were enhanced. NAM was used as an inhibitor for SIRT5. Inhibition of SIRT5 by NAM also further demonstrated that SIRT5 attenuated the oxidative damage caused by H_2_O_2_.

As an antioxidant, the beneficial effects of RES are mainly mediated through activating the SIRT1 pathway in vitro and in vivo studies [63,64]. Yu et al. [46] verify that RES protects cardiomyocytes from oxidative-stress-induced apoptosis by activating SIRT1, SIRT3, SIRT4, and SIRT7. RES could inhibit TNF-α-induced ROS production via activating SIRT1~5 of endothelial cells [46]. RES, through regulating SIRT5 expression and reducing NAD^+^ level, interferes with mitochondrial metabolism and oxidative damage of the hippocampus in elder rats [28]. Our results indicated that RES alleviated intracellular ROS and enhanced multiple antioxidant enzymes in bMECs induced by H_2_O_2_. Inhibition of SIRT5 with NAM weakens the antioxidant capacity of RES and implies that RES resisted oxidative stress associated with SIRT5. Interestingly, we further found that RES activated SIRT5 expression and the binding energy value was −9.0 kcal/mol with SIRT5, suggesting that RES might alleviate oxidative stress by activating SIRT5.

The NADP^+^/NADPH couple is necessary for ensuring cellular redox homeostasis and participating in many biological processes. Oxidative imbalance caused by mitochondrial dysfunction increases the ratio of NADP^+^/NADPH [47,48]. Moreover, NADPH is a cofactor of glutathione reductase in mitochondria, which produces more oxidized states of the glutathione couple. The ratio of GSSG/GSH is also a crucial marker reflecting the redox status. Certainly, documents have suggested the role of SIRTs in regulating antioxidants and redox signaling pathways. The dependence on NAD^+^ for their deacetylase activity allows SIRTs an ideal opportunity to regulate redox reactions via modulating transcription factors that control antioxidant enzymatic expression and cellular NAD^+^/NADH ratios [65]. Research suggests that SIRT5 may promote NADPH production and alleviate cellular ROS levels by activating related enzymes in the tricarboxylic acid cycle [66]. As expected, the elevated NADP^+^/NADPH and declined GSH/GSSG of cells in the H_2_O_2_ group were reversed by an overexpression of SIRT5 or RES treatment. This is probably due to oxidative stress consuming NADPH when cells are treated with H_2_O_2_, while not in SIRT5 overexpression or in RES-treated cells elevating NADPH production. Moreover, inhibiting endogenous SIRT5 could attenuate this effect of RES, suggesting that RES elevated cellular NADPH and GSH production via activating SIRT5. This is of significant importance for confronting external stimulation, keeping cellular environmental homeostasis, and maintaining normal cell function.

NADPH is necessary for the synthesis of ATP by providing reducing equivalents. The knockout of SIRT5 has been especially verified to cause an apparent decrease in ATP synthase enzymatic activity [67]. ROS in mitochondria can change the permeability of mitochondria, accompanied with a low ATP level and high oxygen consumption [68]. In this study, H_2_O_2_ significantly disrupted the morphology of mitochondria, decreased the mitochondrial membrane potential, induced autophagy, and increased the mitochondrial superoxide level. Conversely, the MMP was elevated, mitochondrial morphology was restored, and autophagolysosome and superoxide levels were reduced in the overexpression of SIRT5 cells or RES-treated bMEC exposure to H_2_O_2_. These findings suggested that SIRT5 or RES might attenuate mitochondrial damage in bMECs exposed to H_2_O_2_. Moreover, inhibiting endogenous SIRT5 could alleviate this effect of RES. Mitochondrial dysfunction can affect ATP production. SDH is a marker as evaluating mitochondrial ATP production, and mitochondrial destruction reduces SDH activity [29,30]. Our results showed that H_2_O_2_ notably inhibited ATP synthesis and SDH enzyme activity, and while SIRT5 and RES notably attenuated this reduction, ATP levels and SDH enzyme activity were enhanced. In addition, when NAM was used to inhibit endogenous SIRT5, we found that ATP levels and SDH enzyme activity were further reduced. Furthermore, inhibiting endogenous SIRT5 can weaken the effect of RES on ATP synthesis and SDH enzyme activity. IDH2 participates in antioxidant activity in mitochondria and promotes NADPH and GSH production, which are the main antioxidants for improving ROS damage [35,36]. In this study, RES and SIRT5 significantly enhanced the activity of IDH2. Inhibiting SIRT5 with NAM also confirmed this point. Moreover, RES enhanced IDH2 activity associated with SIRT5. Furthermore, we verified that SDH and IDH2 were among the interaction network of cattle SIRT5, and SIRT5 could form a protein–protein bond with SDHA, especially hydrogen bonds formed with SDHA at K362 site. K362 may be a main site for SIRT5-regulated post-translational modification, and further mass spectrometry and other experimental methods are necessary to verify this point. RES bound to hydrogen bonds in the active pocket of SDHA and IDH2, indicating that there was a strong interaction between RES and SDHA or IDH2. Collectively, RES improved mitochondrial dysfunction via activating SIRT5.

IDH2 as a substrate of SIRT5 reduced the oxidative damage treated with NH_4_Cl, and SIRT5 further desuccinylated IDH2 to elevate its activity, leading to an increase in the NADPH level and alleviating high-level ROS in bMECs induced by ammonia [21]. Hence, we detected the succinylation level of IDH2 and found that RES reduced the intracellular succinylation level of IDH2 by interacting with endogenous IDH2 and catalyzing its desuccinylation, further elevating the enzymatic activity of IDH2 associated with SIRT5. Moreover, inhibiting endogenous SIRT5 could attenuate this effect of RES. Altogether, RES might act as an activator of SIRT5 and enhance IDH2 enzymatic activity via activating the SIRT5-IDH2 axis, resulting in increased GSH and NADPH production, attenuating the oxidative stress of bMECs exposed to H_2_O_2_. These results not only offer new perspectives for the antioxidant defense role of RES in cells, but also provide theoretical evidence for the antioxidant therapy strategies based on SIRT5. Hence, RES may be useful to prevent and control bovine mastitis by relieving oxidative damage.

## 5. Conclusions

In conclusion, RES boosts the antioxidant capacity by activating SIRT5 and elevates intracellular NADPH and GSH levels, as well as ensures mitochondrial function stability of bMECs against H_2_O_2_-induced oxidative damage, as suggested by the abatement in oxidative damage and inflammation. Inhibiting SIRT5 with NAM reduces the ability of RES to oppose oxidative damage. RES promotes IDH2 dessuccinylation and increases IDH2 enzymatic activity via activating SIRT5 (Figure 16). Collectively, these outcomes suggest that RES can function as an activator of SIRT5 to attenuate the oxidative stress of bMECs via activating the SIRT5-IDH2 axis, resulting in increased GSH and NADPH production. Therefore, RES may be useful to prevent and control bovine mastitis by relieving oxidative stress.

However, in vivo studies are crucial for confirming the relevance of in vitro findings in real biological systems. Future efforts should strive to develop relevant animal models to further confirm and expand our current findings, laying a more solid foundation for a comprehensive elucidation of the role of RES in resisting oxidative stress associated with SIRT5. Based on the molecular docking results, we speculate that K362 may be a key site for SIRT5-mediated SDHA post-translational modification. We will conduct further research using mass spectrometry and other related techniques to confirm the role of the K362 site in SIRT5-mediated SDHA post-translational modification.

## Figures and Tables

**Figure 1 antioxidants-14-01171-f001:**
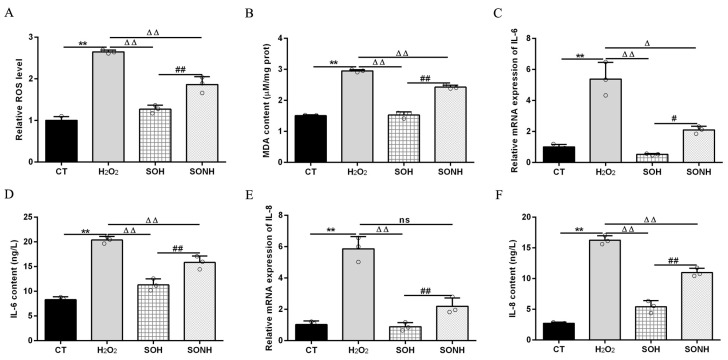
SIRT5 attenuated cellular oxidative damage in bMECs exposed to H_2_O_2_. When bMECs and SIRT5-overexpressed cells were cultured, reaching the confluent extent of 80~90%, they were allocated randomly into four groups. bMECs in the CT group were seeded in a basal medium. bMECs in the H_2_O_2_ group were exposed to 500 µM H_2_O_2_ for 24 h. SIRT5-overexpressed cells in the SOH group were exposed to 500 µM H_2_O_2_ for 24 h. SIRT5-overexpressed cells in the SONH group were exposed to 500 µM H_2_O_2_ and 50 µM NAM for 24 h. ROS level (**A**), MDA content (**B**), and mRNA abundance and content of IL-6 and IL-8 (**C**–**F**). Data are presented in the form of mean ± SEM and repeated at least three times. Two-tailed unpaired *t*-test was executed between two groups; one-way ANOVA was performed among multiple groups. Note: CT: control; H_2_O_2_: 500 μM; SOH: SIRT5-overexpressed cell+H_2_O_2_; SONH: SIRT5-overexpressed cell+H_2_O_2_+NAM. ** *p* < 0.01 vs. CT group; Δ *p* < 0.05, ΔΔ *p* < 0.01 vs. H_2_O_2_ group; # *p* < 0.05, ## *p* < 0.01 vs. SOH group.

**Figure 2 antioxidants-14-01171-f002:**
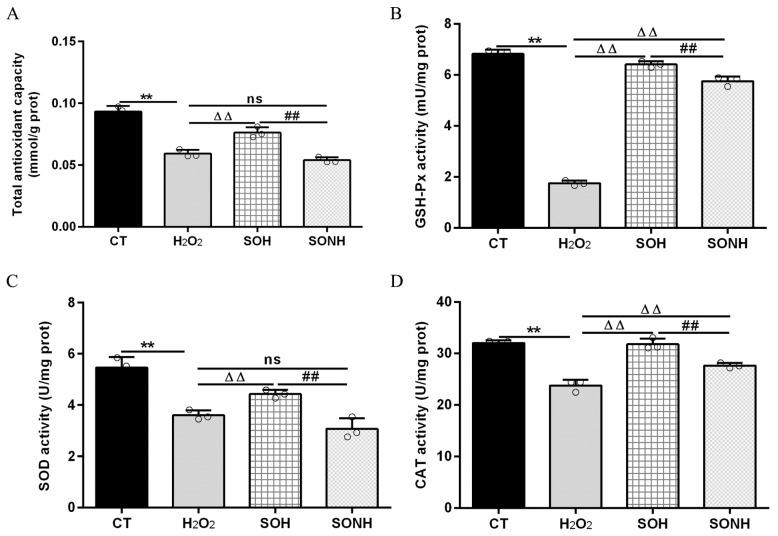
SIRT5 increased the antioxidant capacity of bMECs. When bMECs and SIRT5-overexpressed cells were cultured, reaching the confluent extent of 80~90%, they were allocated randomly into four groups. bMECs in the CT group were seeded in a basal medium. bMECs in the H_2_O_2_ group were exposed to 500 µM H_2_O_2_ for 24 h. SIRT5-overexpressed cells in the SOH group were exposed to 500 µM H_2_O_2_ for 24 h. SIRT5-overexpressed cells in the SONH group were exposed to 500 µM H_2_O_2_ and 50 µM NAM for 24 h. T-AOC content (**A**), GSH-Px activity (**B**), SOD activity (**C**), and CAT activity (**D**). Data are presented in the form of mean ± SEM and repeated at least three times. Two-tailed unpaired *t*-test was executed between two groups; one-way ANOVA was performed among multiple groups. Note: CT: control; H_2_O_2_: 500 μM; SOH: SIRT5-overexpressed cell+H_2_O_2_; SONH: SIRT5-overexpressed cell+H_2_O_2_+NAM. ** *p* < 0.01 vs. CT group, ΔΔ *p* < 0.01 vs. H_2_O_2_ group, ## *p* < 0.01 vs. SOH group.

**Figure 3 antioxidants-14-01171-f003:**
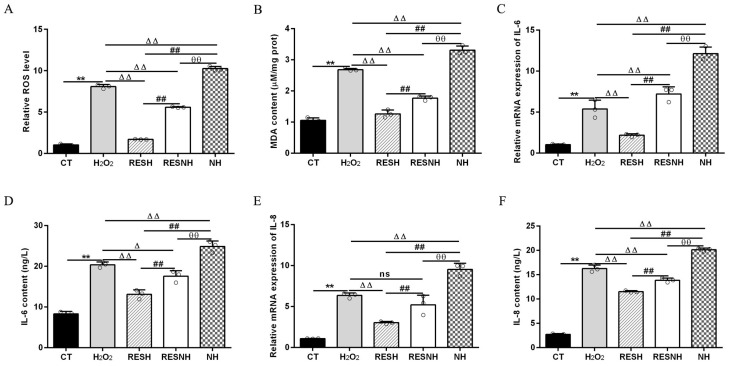
RES ameliorated the oxidative stress associated with SIRT5. When bMECs were cultured, reaching the confluent extent of 80~90%, they were allocated randomly into five groups. bMECs in the CT group were cultured in a basal medium. bMECs in the H_2_O_2_ group were exposed to 500 µM H_2_O_2_ for 24 h. bMECs in the RESH group were exposed to 500 μM H_2_O_2_ and 40 μM RES for 24 h. bMECs in the RESNH group were exposed to 500 µM H_2_O_2_, 40 µM RES, and 50 µM NAM for 24 h. bMECs in the NH group were exposed to 500 µM H_2_O_2_ and 50 µM NAM for 24 h. ROS level (**A**), MDA content (**B**), and mRNA abundance and content of IL-6 and IL-8 (**C**–**F**). Data are presented in the form of mean ± SEM and repeated at least three times. Two-tailed unpaired *t*-test was executed between the two groups; one-way ANOVA was performed among multiple groups. Note: CTs: controls; H_2_O_2_: 500 μM; RESH: H_2_O_2_+RES; RESNH: H_2_O_2_+RES+NAM; NH: H_2_O_2_+NAM. ** *p* < 0.01 vs. CT group; Δ *p* < 0.05, ΔΔ *p* < 0.01 vs. H_2_O_2_ group, ## *p* < 0.01 vs. RESH group; θθ *p* < 0.01 vs. RESNH group.

**Figure 4 antioxidants-14-01171-f004:**
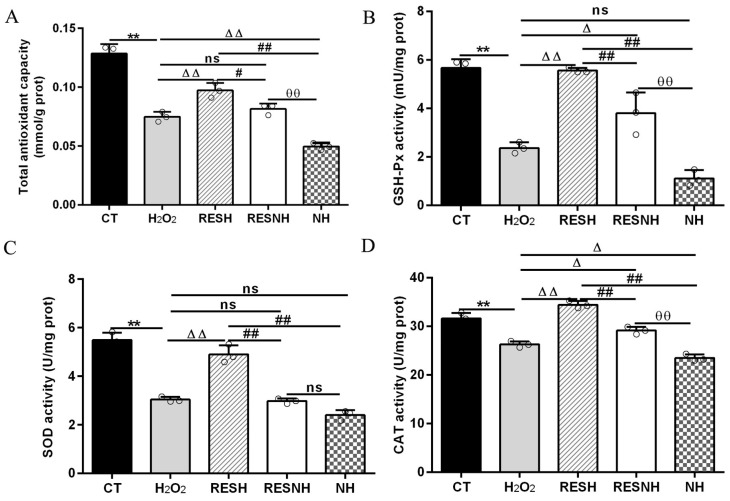
RES enhanced the cellular antioxidant capacity associated with SIRT5. When bMECs were cultured, reaching the confluent extent of 80~90%, they were allocated randomly into five groups. bMECs in the CT group were cultured in a basal medium. bMECs in the H_2_O_2_ group were exposed to 500 µM H_2_O_2_ for 24 h. bMECs in the RESH group were exposed to 500 μM H_2_O_2_ and 40 μM RES for 24 h. bMECs in the RESNH group were exposed to 500 µM H_2_O_2_, 40 µM RES, and 50 µM NAM for 24 h. bMECs in the NH group were exposed to 500 µM H_2_O_2_ and 50 µM NAM for 24 h. T-AOC content (**A**), GSH-Px activity (**B**), SOD activity (**C**), and CAT activity (**D**). Data are presented in the form of mean ± SEM and repeated at least three times. Two-tailed unpaired *t*-test was executed between two groups; one-way ANOVA was performed among multiple groups. Note: CTs: controls; H_2_O_2_: 500 μM; RESH: H_2_O_2_+RES; RESNH: H_2_O_2_+RES+NAM; NH: H_2_O_2_+NAM. ** *p* < 0.01 vs. CT group; Δ *p* < 0.05, ΔΔ *p* < 0.01 vs. H_2_O_2_ group; # *p* < 0.01, ## *p* < 0.01 vs. RESH group; θθ *p* < 0.01 vs. RESNH group.

**Figure 5 antioxidants-14-01171-f005:**
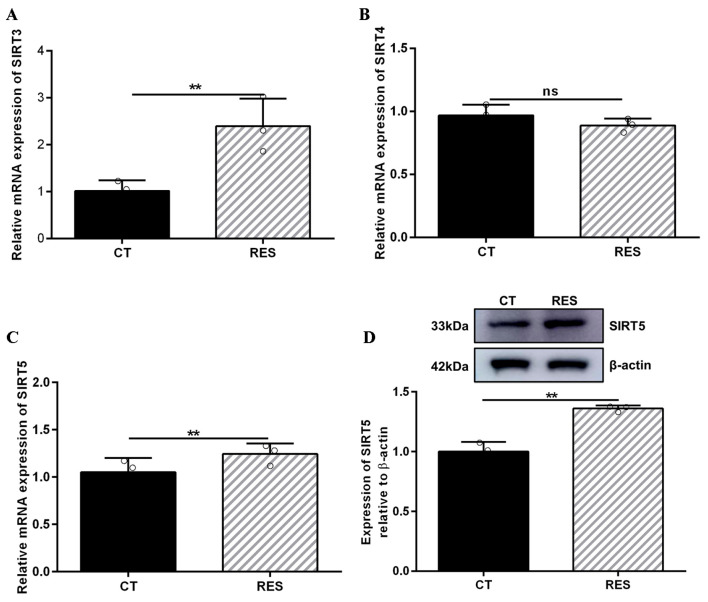
Effect of RES on SIRT expression in bMECs. When bMECs were cultured, reaching the confluent extent of 80~90%, they were allocated randomly into two groups. bMECs in the CT group were cultured in a basal medium. bMECs in the RES group were handled with 40 µM RES for 24 h. SIRT3 mRNA abundance in bMECs (**A**), SIRT4 mRNA abundance in bMECs (**B**), SIRT5 mRNA abundance in bMECs (**C**), and the protein expression of SIRT5 in bMECs (**D**). Data are presented in the form of mean ± SEM and repeated at least three times. Two-tailed unpaired *t*-test was executed between two groups. ** *p* < 0.01 vs. CT group.

**Figure 6 antioxidants-14-01171-f006:**
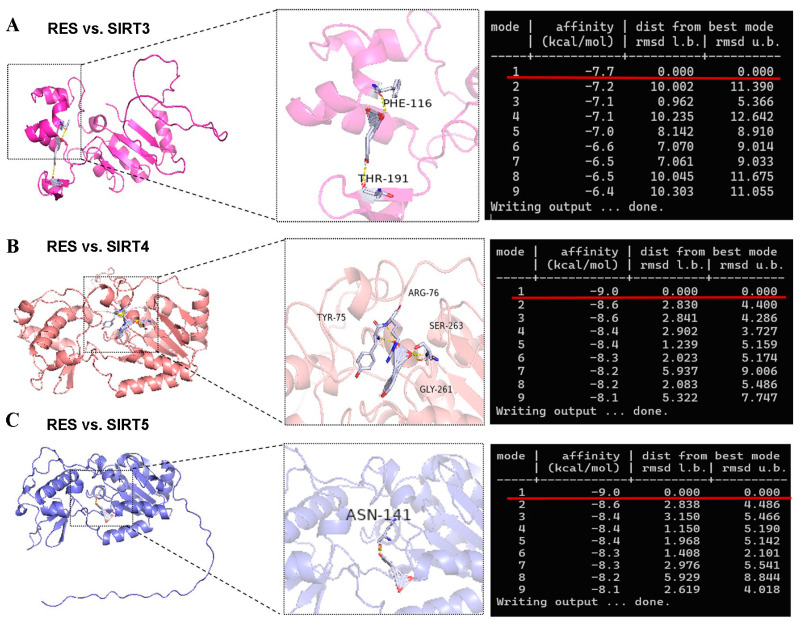
Visualization of molecular docking between RES and SIRTs. The AutoDock Vina software is used for molecular docking to obtain binding energy. The results of molecular docking are visualized and analyzed using PyMOL software to obtain a 3D conformation map. The molecular docking structure and binding energy of RES and SIRT3 (**A**) or SIRT4 (**B**) or SIRT5 (**C**).

**Figure 7 antioxidants-14-01171-f007:**
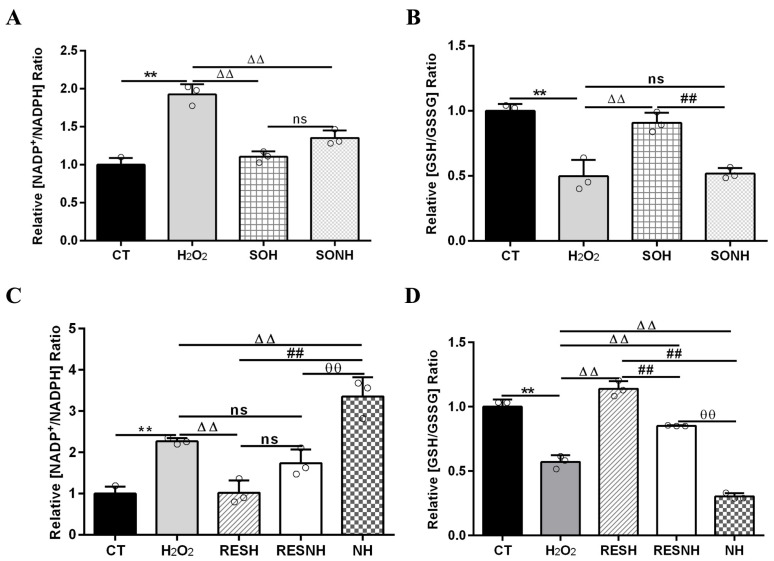
RES elevated NADPH and GSH contents via activating SIRT5. When bMECs and SIRT5-overexpressed cells were cultured, reaching the confluent extent of 80~90%, they were randomly divided into seven groups. bMECs in the CT group were cultured in a basal medium. bMECs in the H_2_O_2_ group were treated with 500 µM H_2_O_2_ for 24 h. bMECs in the RESH group were handled with 500 μM H_2_O_2_ and 40 μM RES for 24 h. bMECs in the RESNH group were handled with 500 µM H_2_O_2_, 40 µM RES, and 50 µM NAM for 24 h. bMECs in the NH group were treated with 500 µM H_2_O_2_ and 50 µM NAM for 24 h. SIRT5-overexpressed cells in the SOH group were handled with 500 µM H_2_O_2_ for 24 h. SIRT5-overexpressed cells in the SONH group were treated with 500 µM H_2_O_2_ and 50 µM NAM for 24 h. NADP^+^/NADPH (**A**,**C**) and GSH/GSSG (**B**,**D**). Data are presented in the form of mean ± SEM and repeated at least three times. Two-tailed unpaired *t*-test was executed between two groups; one-way ANOVA was performed among multiple groups. Note: CTs: controls; H_2_O_2_: 500 μM; SOH: SIRT5-overexpressed cell+H_2_O_2_; SONH: SIRT5-overexpressed cell+H_2_O_2_+NAM; RESH: H_2_O_2_+RES; RESNH: H_2_O_2_+RES+NAM; NH: H_2_O_2_+NAM. ** *p* < 0.01 vs. CT group, ΔΔ *p* < 0.01 vs. H_2_O_2_ group, ## *p* < 0.01 vs. SOH group or RESH group, θθ *p* < 0.01 vs. RESNH group.

**Figure 8 antioxidants-14-01171-f008:**
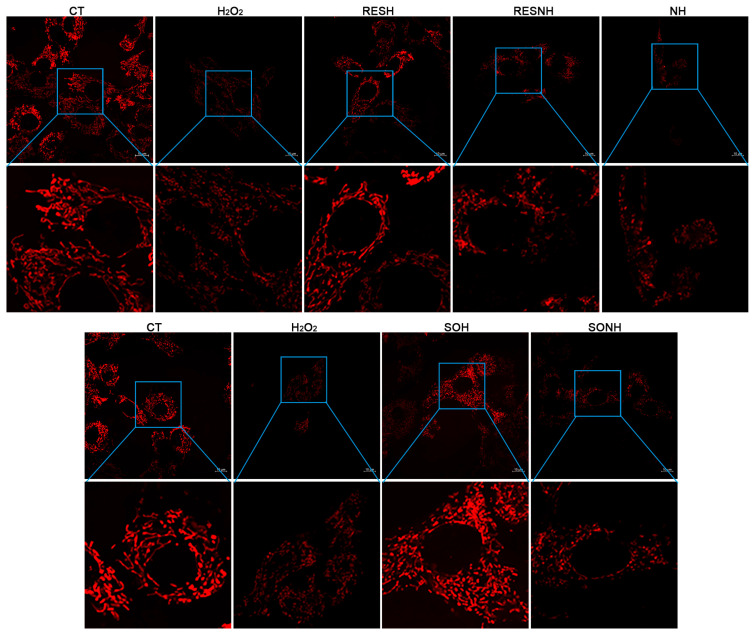
RES improved mitochondrial morphology via activating SIRT5. bMECs and SIRT5-overexpressed cells were cultured, reaching the confluent extent of 80~90%, and they were randomly divided into seven groups. bMECs in the CT group were cultured in a basal medium. bMECs in the H_2_O_2_ group were treated with 500 µM H_2_O_2_ for 24 h. bMECs in the RESH group were handled with 500 μM H_2_O_2_ and 40 μM RES for 24 h. bMECs in the RESNH group were treated with 500 µM H_2_O_2_, 40 µM RES, and 50 µM NAM for 24 h. bMECs in the NH group were handled with 500 µM H_2_O_2_ and 50 µM NAM for 24 h. SIRT5-overexpressed cells in SOH group were handled with 500 µM H_2_O_2_ for 24 h. SIRT5-overexpressed cells in the SONH group were treated with 500 µM H_2_O_2_ and 50 µM NAM for 24 h. Cells processed as the above description were marked with the fluorescent probe Mito Tracker red CMXRos and mitochondrial morphological changes were observed by confocal microscopy. Scale bars: 10 μm. Note: CTs: controls; H_2_O_2_: 500 μM; SOH: SIRT5-overexpressed cell+H_2_O_2_; SONH: SIRT5-overexpressed cell+H_2_O_2_+NAM; RESH: H_2_O_2_+RES; RESNH: H_2_O_2_+RES+NAM; NH: H_2_O_2_+NAM.

**Figure 9 antioxidants-14-01171-f009:**
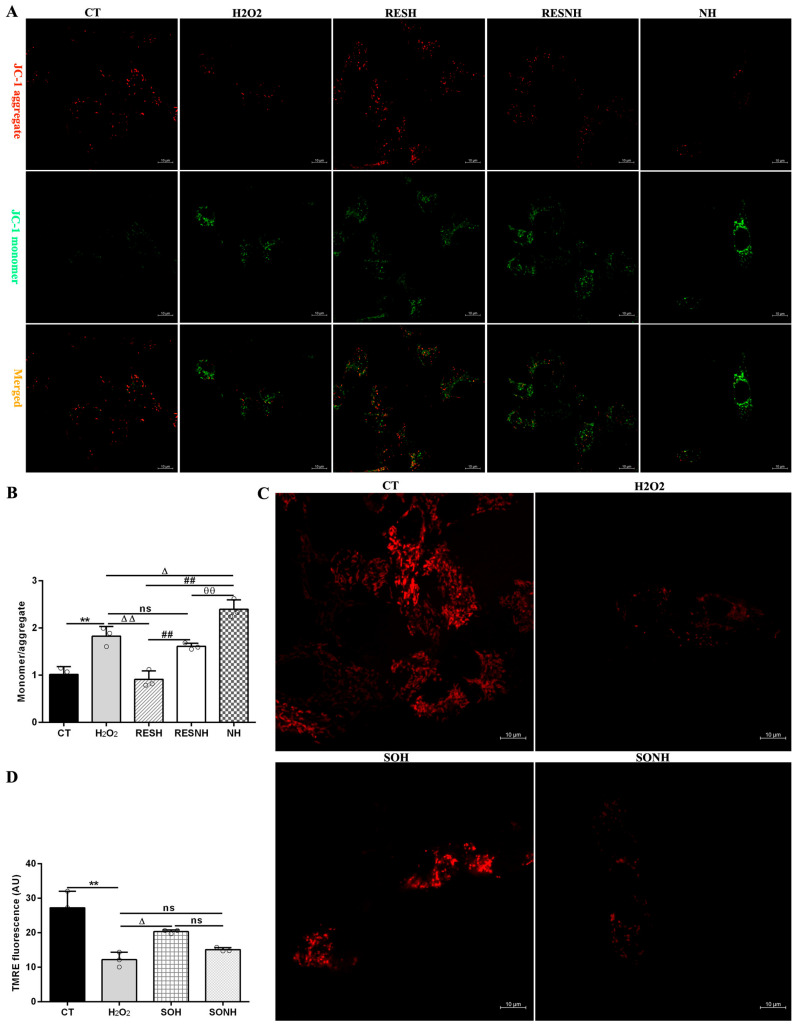
RES enhanced the mitochondrial membrane potential via activating SIRT5. bMECs and SIRT5-overexpressed cells were cultured, reaching the confluent extent of 80~90%, and they were randomly divided into seven groups. bMECs in the CT group were cultured in a basal medium. bMECs in the H_2_O_2_ group were treated with 500 µM H_2_O_2_ for 24 h. bMECs in the RESH group were treated with 500 μM H_2_O_2_ and 40 μM RES for 24 h. bMECs in the RESNH group were handled with 500 µM H_2_O_2_, 40 µM RES, and 50 µM NAM for 24 h. bMECs in the NH group were handled with 500 µM H_2_O_2_ and 50 µM NAM for 24 h. SIRT5-overexpressed cells in the SOH group were handled with 500 µM H_2_O_2_ for 24 h. SIRT5-overexpressed cells in the SONH group were treated with 500 µM H_2_O_2_ and 50 µM NAM for 24 h. (**A**) The above cells were marked with JC-1 fluorescent probe to detect MMP levels under confocal microscopy. JC-1 monomer: red; JC-1 aggregate: green. Scale bars: 10 μm. (**B**) Quantitative ratio of red/green fluorescence intensity. (**C**) TMRE detection of MMP. Scale bars: 10 μm. (**D**) Image J software was used to quantify the fluorescence intensity. Data are presented in the form of mean ± SEM and repeated at least three times. Two-tailed unpaired *t*-test was executed between the two groups; one-way ANOVA was performed among multiple groups. Note: CTs: controls; H_2_O_2_: 500 μM; SOH: SIRT5-overexpressed cell+H_2_O_2_; SONH: SIRT5-overexpressed cell+H_2_O_2_+NAM; RESH: H_2_O_2_+RES; RESNH: H_2_O_2_+RES+NAM; NH: H_2_O_2_+NAM. ** *p* < 0.01 vs. the CT group; Δ *p* < 0.05, ΔΔ *p* < 0.01 vs. H_2_O_2_ group; ## *p* < 0.01 vs. the SOH group or RESH group; θθ *p* < 0.01 vs. the RESNH group.

**Figure 10 antioxidants-14-01171-f010:**
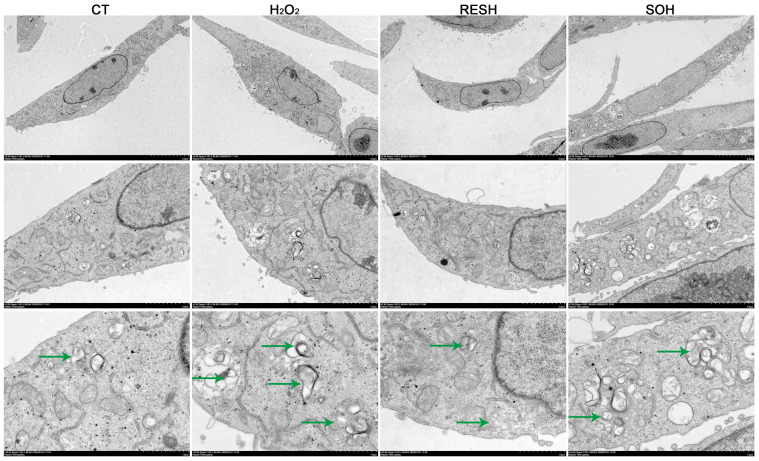
Effect of RES and SIRT5 on the autophagy vesicle of bMECs by transmission electron microscopy observation. bMECs and SIRT5-overexpressed cells were cultured reaching the confluent extent of 80~90%, and they were allocated randomly into four groups. bMECs in the CT group were cultured in a basal medium. bMECs in the H_2_O_2_ group were dealt with 500 µM H_2_O_2_ for 24 h. bMECs in the RESH group were dealt with 500 μM H_2_O_2_ and 40 μM RES for 24 h. SIRT5-overexpressed cells in the SOH group were dealt with 500 µM H_2_O_2_ for 24 h. Scale bars: 5 μm (**upper**), 2 μm (**middle row**) and 1 μm (**bottom**), the green arrows point to the autolysosomes. Note: CTs: controls; H_2_O_2_: 500 μM; RESH: H_2_O_2_+RES; SOH: SIRT5-overexpressed cell+H_2_O_2_.

**Figure 11 antioxidants-14-01171-f011:**
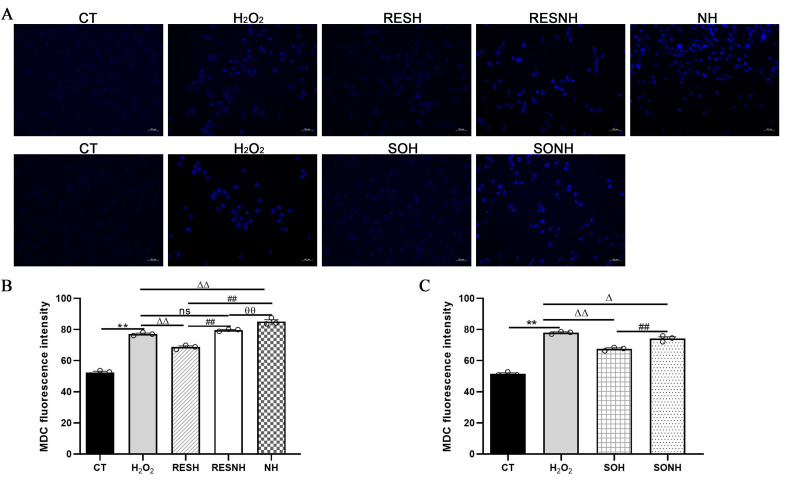
Labeling of the autophagic vesicle structure by MDC. bMECs and SIRT5-overexpressed cells were cultured reaching the confluent extent of 80~90%, and they were randomly divided into seven groups. bMECs in the CT group were cultured in a basal medium. bMECs in the H_2_O_2_ group were dealt with 500 µM H_2_O_2_ for 24 h. bMECs in the RESH group were handled with 500 μM H_2_O_2_ and 40 μM RES for 24 h. bMECs in the RESNH group were treated with 500 µM H_2_O_2_, 40 µM RES, and 50 µM NAM for 24 h. bMECs in the NH group were handled with 500 µM H_2_O_2_ and 50 µM NAM for 24 h. SIRT5-overexpressed cells in SOH group were treated with 500 µM H_2_O_2_ for 24 h. SIRT5-overexpressed cells in the SONH group were handled with 500 µM H_2_O_2_ and 50 µM NAM for 24 h. (**A**) Autophagic vesicle structure by MDC detection. Scale bars: 50 μm. (**B**,**C**) Fluorescence intensity was analyzed by ImageJ. Data are presented in the form of mean ± SEM and repeated at least three times. Two-tailed unpaired *t*-test was executed between the two groups; one-way ANOVA was performed among multiple groups. Note: CTs: controls; H_2_O_2_: 500 μM; SOH: SIRT5-overexpressed cell+H_2_O_2_; SONH: SIRT5-overexpressed cell+H_2_O_2_+NAM; RESH: H_2_O_2_+RES; RESNH: H_2_O_2_+RES+NAM; NH: H_2_O_2_+NAM. ** *p* < 0.01 vs. CT group; Δ *p* < 0.05, ΔΔ *p* < 0.01 vs. H_2_O_2_ group; ## *p* < 0.01 vs. the SOH group or the RESH group; θθ *p* < 0.01 vs. the RESNH group.

**Figure 12 antioxidants-14-01171-f012:**
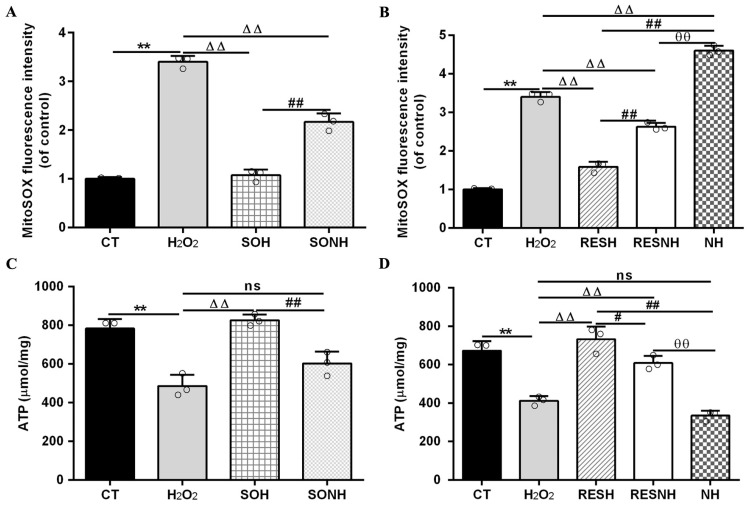
Effect of RES and SIRT5 on mitochondrial superoxide and ATP content. bMECs and SIRT5-overexpressed cells were cultured, reaching the confluent extent of 80~90%, and they were randomly divided into seven groups. bMECs in the CT group were cultured in a basal medium. bMECs in the H_2_O_2_ group were treated with 500 µM H_2_O_2_ for 24 h. bMECs in the RESH group were treated with 500 μM H_2_O_2_ and 40 μM RES for 24 h. bMECs in the RESNH group were treated with 500 µM H_2_O_2_, 40 µM RES, and 50 µM NAM for 24 h. bMECs in the NH group were treated with 500 µM H_2_O_2_ and 50 µM NAM for 24 h. SIRT5-overexpressed cells in the SOH group were treated with 500 µM H_2_O_2_ for 24 h. SIRT5-overexpressed cells in the SONH group were treated with 500 µM H_2_O_2_ and 50 µM NAM for 24 h. (**A**,**B**) Mitochondrial superoxide production; (**C**,**D**) ATP content. Data are presented in the form of mean ± SEM and repeated at least three times. Two-tailed unpaired *t*-test was executed between the two groups; one-way ANOVA was performed among multiple groups. Note: CTs: controls; H_2_O_2_: 500 μM; SOH: SIRT5-overexpressed cell+H_2_O_2_; SONH: SIRT5-overexpressed cell+H_2_O_2_+NAM; RESH: H_2_O_2_+RES; RESNH: H_2_O_2_+RES+NAM; NH: H_2_O_2_+NAM. ** *p* < 0.01 vs. the CT group; ΔΔ *p* < 0.01 vs. the H_2_O_2_ group; # *p* < 0.05, ## *p* < 0.01 vs. the SOH group or RESH group; θθ *p* < 0.01 vs. the RESNH group.

**Figure 13 antioxidants-14-01171-f013:**
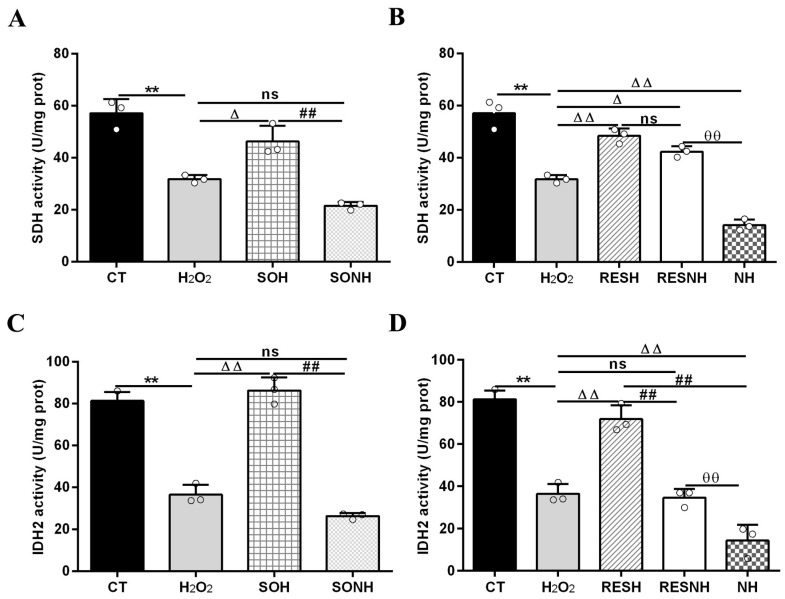
Effects of RES and SIRT5 on SDH and IDH2 enzymatic activities. bMECs and SIRT5-overexpressed cells were cultured reaching the confluent extent of 80~90%, and they were randomly divided into seven groups. bMECs in the CT group were cultured in a basal medium. bMECs in the H_2_O_2_ group were treated with 500 µM H_2_O_2_ for 24 h. bMECs in the RESH group were handled with 500 μM H_2_O_2_ and 40 μM RES for 24 h. bMECs in the RESNH group were treated with 500 µM H_2_O_2_, 40 µM RES, and 50 µM NAM for 24 h. bMECs in the NH group were treated with 500 µM H_2_O_2_ and 50 µM NAM for 24 h. SIRT5-overexpressed cells in SOH group were treated with 500 µM H_2_O_2_ for 24 h. SIRT5-overexpressed cells in the SONH group were treated with 500 µM H_2_O_2_ and 50 µM NAM for 24 h. (**A**,**B**) SDH activity; (**C**,**D**) IDH2 activity. Data are presented in the form of mean ± SEM and repeated at least three times. Two-tailed unpaired *t*-test was executed between the two groups; one-way ANOVA was performed among multiple groups. Note: CTs: controls; H_2_O_2_: 500 μM; SOH: SIRT5-overexpressed cell+H_2_O_2_; SONH: SIRT5-overexpressed cell+H_2_O_2_+NAM; RESH: H_2_O_2_+RES; RESNH: H_2_O_2_+RES+NAM; NH: H_2_O_2_+NAM. ** *p* < 0.01 vs. the CT group; Δ *p* < 0.05, ΔΔ *p* < 0.01 vs. the H_2_O_2_ group; ## *p* < 0.01 vs. the SOH group or RESH group; θθ *p* < 0.01 vs. the RESNH group.

**Figure 14 antioxidants-14-01171-f014:**
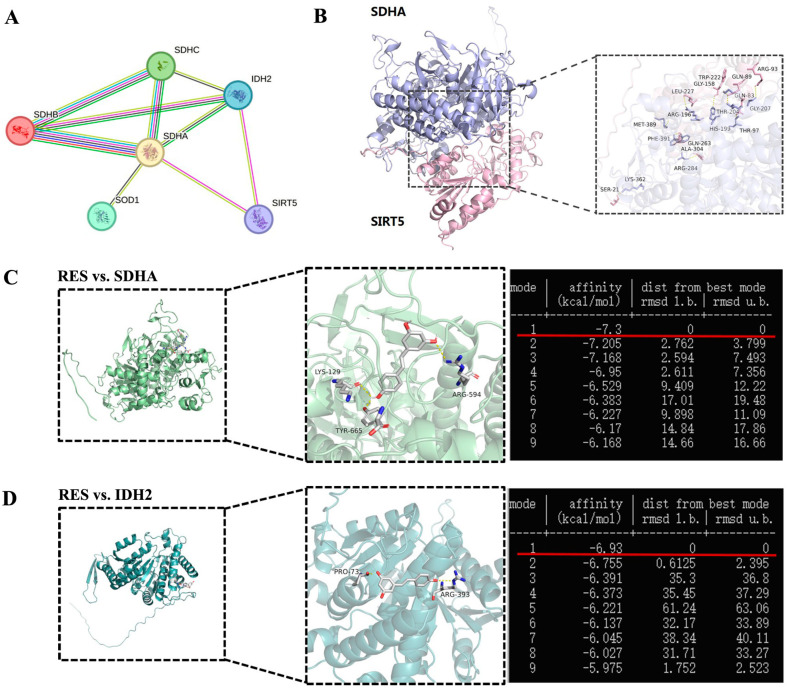
Visualization of molecular docking between SIRT5 and SDHA or RES and SDHA or IDH2. (**A**) The interaction networks among SDHA, IDH2, and SIRT5. Each node represents a related gene. (**B**) Molecular docking structure of SIRT5 and SDHA. (**C**) Molecular docking structure and binding energy of RES and SDHA. (**D**) Molecular docking structure and binding energy of RES and IDH2.

**Figure 15 antioxidants-14-01171-f015:**
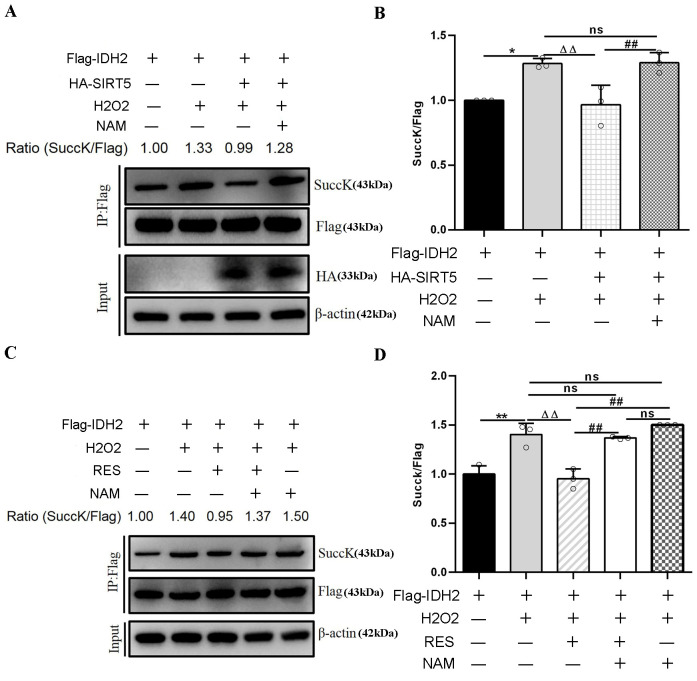
Effects of RES on the succinylation level of IDH2. HEK293T cells were transfected with Flag-IDH2 plasmid or HA-SIRT5 and FlagIDH2 plasmids and treated with 40 µM RES and 50 µM NAM with or without 500 µM H_2_O_2_ for 24 h. In HEK293T cells, purification of the Flag-IDH2 protein is performed using IP beads and Western blotting to detect its succinylation level. (**A**,**C**) Immunoprecipitation and Western blotting were executed to evaluate the succinylation level of IDH2 using succinyl-lysine-specific antibodies. (**B**,**D**) Relative intensity of SuccK/Flag. Data are presented in the form of mean ± SEM and repeated at least three times. Two-tailed unpaired *t*-test was executed between the two groups; one-way ANOVA was performed among multiple groups. * *p* < 0.05, ** *p* < 0.01 vs. the CT group; ΔΔ *p* < 0.01 vs. the H_2_O_2_ group; ## *p* < 0.01 vs. the SOH group or RESH group.

**Figure 16 antioxidants-14-01171-f016:**
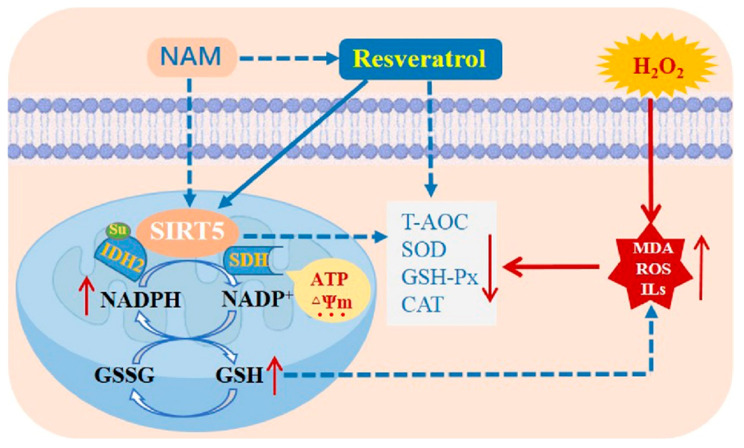
A model of RES alleviates the oxidative stress of bMECs via activating the SIRT5-IDH2 axis. RES boosts the antioxidant capacity by activating SIRT5 and increases intracellular NADPH and GSH levels, as well as guarantees mitochondrial function stability of bMECs against H_2_O_2_-induced oxidative stress, as indicated by the abatement in oxidative damage and inflammation. Inhibiting SIRT5 with NAM attenuates the ability of RES to resist oxidative stress. RES promotes IDH2 dessuccinylation and increases IDH2 enzymatic activity via activating SIRT5. This study has demonstrated that RES can function as an activator of SIRT5 to attenuate the oxidative stress of bMECs via activating the SIRT5-IDH2 axis, resulting in increased GSH and NADPH production. Note: the dashed line represents the inhibition role, and the solid line represents the promotion role. Note: The dashed line denotes inhibition, and the solid line denotes promotion.

**Table 1 antioxidants-14-01171-t001:** Primer information used in this research.

Target Genes	Sequence (5′–3′)	Accession No.
*IL-6*	F: AGACTACTTCTGACCACTCCA R: GCTGCTTTCACACTCATCATTC	NM_173923.2
*IL-8*	F: TGAGTACAGAACTTCGATGCC R: GTGTGGCCCACTCTCAATAA	NM_173925.2
*SIRT3*	F: CCGCTGGCCTCGTATTCC R: TCTGGCAGGCTCTGGTCTTA	NM_001206669.1
*SIRT4*	F: GGGATCATCCTTGCAGGTGTA R: CAGAGATGCCAGGTCATCGG	NM_001075785.1
*SIRT5*	F: TTGTGGAGTTGTGGCTGAGA R: GTCCCCACCACTAGACACAG	NM_001034295.2
*β-actin*	F: GATATTGCTGCGCTCGTGG R: GTCAGGATGCCTCTCTTGCT	NM_173979.3

## Data Availability

The datasets used and/or analyzed during the present study are available from the corresponding author upon request.

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
