# Peer review of "Resveratrol Alleviated Oxidative Damage of Bovine Mammary Epithelial Cells via Activating SIRT5-IDH2 Axis"

_antioxidants, 2025, doi:10.3390/antiox14101171_

Round 1

Reviewer 1 Report

This paper, author suggested that Resveratrol (RES) protects bMECs from oxidative stress by activating SIRT5.
This enhances mitochondrial function and boosts NADPH and GSH. Thus, RES may help prevent bovine mastitis.
Resaerch is very interesting and the flow of the stud is smooth. however, addiational results need to be included at certain points to clarify key aspects. 

In figure2. SOH and SONH groups, could you mark H2O2 treatment in figure. It looks like only H2O2 treated to one group although writing in figure legend

there is lots of experimental group. it is not eay to notice. could you reorganize it 
I think it would be better to intutitively put one in the figure as shown below 

CT: controls
H2O2: 500uM 
RESH: H2O2+RES
RESNH: H2O2+RES+NAM 
NH: H2O2+ NAM 
SOH: SIRT 5 overexpression cell + H2O2
SOHN: SIRT 5 overexpression cell + H2O2+ NAM

Figure 5, the graph is too thick and the immunoblot image is too small. match the thickness of the other bar graphs 
Why is Sirt3 absent in protein expression images? Sirt3 is also significant in Figure 5A 
why did the author not consider futher research on Sirt3 ? was it based on the molecular docking results (predict)? 

In figure 9. Generally, JC-1 assay, to obatain accurate qunatitiave expression, we add the FACS method. pelase include it if possible 

In figure 10. TEM image, quantitative comparsiom of autophagy is not possible in this figure, gene expression data or protein expression data for detecting autophagy neeed

Author Response

Comment 1

1. In figure2. SOH and SONH groups, could you mark H2O2 treatment in figure. It looks like only H2O2 treated to one group although writing in figure legend. there is lots of experimental group. it is not eay to notice. could you reorganize it 
I think it would be better to intutitively put one in the figure as shown below 

CT: controls
H2O2: 500uM 
RESH: H2O2+RES
RESNH: H2O2+RES+NAM 
NH: H2O2+ NAM 
SOH: SIRT5 overexpression cell + H2O2
SOHN: SIRT5 overexpression cell + H2O2+ NAM

Response 1: We thank you for your careful review and helpful comments on our manuscript. We also appreciate your time and expertise in providing constructive feedback to improve the quality of our work. Considering the size and aesthetics of the images, we have added these details in the legend.

2. Figure 5, the graph is too thick and the immunoblot image is too small. match the thickness of the other bar graphs 
Why is Sirt3 absent in protein expression images? Sirt3 is also significant in Figure 5A 
why did the author not consider futher research on Sirt3 ? was it based on the molecular docking results (predict)? 

Response 2: We thank you for your careful review and helpful comments on our manuscript. We also appreciate your time and expertise in providing constructive feedback to improve the quality of our work. We have redrawn Figure 5. This paper mainly focuses on SIRT5. SIRT3 is only used as a reference and does not detect protein expression. Thank you for your constructive suggestion. Based on the molecular docking results, we will further study SIRT3 in the future.

3. In figure 9. Generally, JC-1 assay, to obatain accurate qunatitiave expression, we add the FACS method. pelase include it if possible 

Response 3: We thank you for your careful review and helpful comments on our manuscript. We also appreciate your time and expertise in providing constructive feedback to improve the quality of our work. Thank you for your constructive suggestion. At present, we are unable to use FACS to detect mitochondrial membrane potential.

4. In figure 10. TEM image, quantitative comparsiom of autophagy is not possible in this figure, gene expression data or protein expression data for detecting autophagy neeed

Response 4: We thank you for your careful review and helpful comments on our manuscript. We also appreciate your time and expertise in providing constructive feedback to improve the quality of our work. Thank you for your constructive suggestion. Afterwards, we will use RT-PCR and WB to detect the expression of autophagy proteins.

Comment 2

1. The abstract needs to be rewritten; it missing a purpose.

Response 1: We thank you for your careful review and helpful comments on our manuscript. We also appreciate your time and expertise in providing constructive feedback to improve the quality of our work. We add this section in the abstract.

2. The introduction is described correctly, but at the end the authors should add the purpose of the research.

Response 2: We thank you for your careful review and helpful comments on our manuscript. We also appreciate your time and expertise in providing constructive feedback to improve the quality of our work. We add this section in the introduction.

3. The material and research methods are described correctly. Modern research methods at the cellular level were used.

Response 3: We thank you for your careful review and helpful comments on our manuscript. We also appreciate your time and expertise in providing constructive feedback to improve the quality of our work.

4. The results are described correctly and well documented with illustrations.

Response 4: We thank you for your careful review and helpful comments on our manuscript. We also appreciate your time and expertise in providing constructive feedback to improve the quality of our work.

5. The discussion needs to be rewritten, as the research was conducted on a cell model, and the results should also be related to veterinary aspects (mastitis conditions).

Response 5: We thank you for your careful review and helpful comments on our manuscript. We also appreciate your time and expertise in providing constructive feedback to improve the quality of our work. We have revised this section.

6. The research provided information on the molecular mechanism at the cellular level.

Response 6: We thank you for your careful review and helpful comments on our manuscript.

7. The discussion should be supplemented with information on the practical aspects of the research conducted.

Response 7: We thank you for your careful review and helpful comments on our manuscript. We have supplemented this section.

8. The research is not very advanced in terms of the techniques used, while the number of authors of the publication is large (is this necessary?).

Response 8: We thank you for your careful review and helpful comments on our manuscript. Because each author has made varying degrees of contribution, their names should be appeared in the article.

Reviewer 2 Report

  1. The abstract needs to be rewritten; it missing a purpose.
  2. The introduction is described correctly, but at the end the authors should add the purpose of the research.
  3. The material and research methods are described correctly. Modern research methods at the cellular level were used.
  4. The results are described correctly and well documented with illustrations.
  5. The discussion needs to be rewritten, as the research was conducted on a cell model, and the results should also be related to veterinary aspects (mastitis conditions).
  6. The research provided information on the molecular mechanism at the cellular level.
  7. The discussion should be supplemented with information on the practical aspects of the research conducted.
  8. The research is not very advanced in terms of the techniques used, while the number of authors of the publication is large (is this necessary?).
  1. The abstract needs to be rewritten; it missing a purpose.
  2. The introduction is described correctly, but at the end the authors should add the purpose of the research.
  3. The material and research methods are described correctly. Modern research methods at the cellular level were used.
  4. The results are described correctly and well documented with illustrations.
  5. The discussion needs to be rewritten, as the research was conducted on a cell model, and the results should also be related to veterinary aspects (mastitis conditions).
  6. The research provided information on the molecular mechanism at the cellular level.
  7. The discussion should be supplemented with information on the practical aspects of the research conducted.
  8. The research is not very advanced in terms of the techniques used, while the number of authors of the publication is large (is this necessary?).

Author Response

(The authors gave the same response as above.)

Round 2

Reviewer 1 Report

The author has not conducted any further experiments on the question, but the current version is better
than the previous one.

The author has not conducted any further experiments on the question, but the current version is better
than the previous one.

Reviewer 2 Report

I accept the additions made. The manuscript has been corrected in accordance with the reviewer's suggestions.

I accept the additions made. The manuscript has been corrected in accordance with the reviewer's suggestions.